# A novel mouse model of obstructive sleep apnea by bulking agent-induced tongue enlargement results in left ventricular contractile dysfunction

**Simon Lebek**[ID]**, Philipp Hegner**[ID]**, Christian Schach, Kathrin Reuthner, Maria Tafelmeier, Lars Siegfried Maier, Michael Arzt, Stefan Wagner** *

Department of Internal Medicine II, University Hospital Regensburg, Regensburg, Germany

* stefan.wagner@ukr.de

## Abstract

### Aims

Obstructive sleep apnea (OSA) is a widespread disease with high global socio-economic impact. However, detailed pathomechanisms are still unclear, partly because current animal models of OSA do not simulate spontaneous airway obstruction. We tested whether polytetrafluoroethylene (PTFE) injection into the tongue induces spontaneous obstructive apneas.

### Methods and results

PTFE (100 µl) was injected into the tongue of 31 male C57BL/6 mice and 28 mice were used as control. Spontaneous apneas and inspiratory flow limitations were recorded by whole-body plethysmography and mRNA expression of the hypoxia marker KDM6A was quantified by qPCR. Left ventricular function was assessed by echocardiography and ventricular CaMKII expression was measured by Western blotting. After PTFE injection, mice showed features of OSA such as significantly increased tongue diameters that were associated with significantly and sustained increased frequencies of inspiratory flow limitations and apneas. Decreased KDM6A mRNA levels indicated chronic hypoxemia. 8 weeks after surgery, PTFE-treated mice showed a significantly reduced left ventricular ejection fraction. Moreover, the severity of diastolic dysfunction (measured as E/e') correlated significantly with the frequency of apneas. Accordingly, CaMKII expression was significantly increased in PTFE mice and correlated significantly with the frequency of apneas.

### Conclusions

We describe here the first mouse model of spontaneous inspiratory flow limitations, obstructive apneas, and hypoxia by tongue enlargement due to PTFE injection. These mice develop systolic and diastolic dysfunction and increased CaMKII expression. This mouse model offers great opportunities to investigate the effects of obstructive apneas.

**Data Availability Statement:** All relevant data are within the manuscript and its Supporting Information files.

**Funding:** SL was funded by the Max Weber scholarship. SW was funded by DFG grants WA 2539/4-1, 5-1, 7-1, and 8-1. LSM was funded by DFG grants MA 1982/5-1 and 7-1. SW and LSM were also funded by the DFG SFB 1350 grant (Project Number 387509280, TPA6), and were supported by the ReForM C program of the faculty. The funders had no role in study design, data collection and analysis, decision to publish, or preparation of the manuscript.

**Competing interests:** MA received grant support from ResMed, the ResMed Foundation, and Philips Respironics as well as lecture and consulting fees from ResMed, Philips Respironics, Boehringer-Ingelheim, NRI, Novartis and Bresotec. There are no other competing interests to declare. This does not alter our adherence to PLOS ONE policies on sharing data and materials.

**Abbreviations:** BW, body weight; CaMKII, Ca/calmodulin-dependent protein kinase II; CIH, chronic intermittent hypoxemia; HIF1α, hypoxia-inducible factor 1α; HW, heart weight; IFL, inspiratory flow limitation; KDM6A, lysine (K)-specific demethylase 6A; LW, lung weight; OSA, obstructive sleep apnea; PTFE, polytetrafluoroethylene.

## Introduction

Obstructive sleep apnea (OSA) is a very common disease with high global socio-economic impact, since almost one billion people are affected worldwide [1]. It is frequently associated with heart failure, cardiac hypertrophy or atrial and ventricular arrhythmias [2–4]. Patients with OSA have been shown to develop worse outcome after myocardial infarction [5]. While treatment with ventilation-therapy may reduce apnea events, not all patients can tolerate it [6] and this treatment may even be harmful for selected patients (e.g. for patients with predominant central apneas) [7]. Thus, the development of novel and more specific treatment concepts is warranted. We have shown that the activity of Ca/Calmodulin-dependent protein kinase II (CaMKII), a key player in heart failure development [8, 9], was increased in the hearts of patients with sleep-disordered breathing, leading to contractile dysfunction and arrythmias [4]. Proposed mechanisms of CaMKII activation in OSA include increased transmural pressure gradients with increased atrial wall stress, increased sympathetic and parasympathetic activation, and reactive oxygen species (ROS) [3, 4, 10, 11]. Unfortunately, patient studies are not always suited to discriminate between different pathophysiological factors. Moreover, the analysis of heterogenous patient populations with comorbidities makes it difficult to interpret the data.

To avoid these pitfalls and to investigate OSA-dependent effects in the absence of potential confounders, appropriate animal models are urgently warranted. However, current animal models of OSA and sleep-disordered breathing in general struggle with many limitations including lack of airway obstruction (intermittent hypoxia models), artificial sedation, lack of availability in small rodents (e.g. mice) for investigation of transgenic animals [12–17].

Interestingly, New Zealand Obese mice have notable thicker tongues with a consecutive increased occurrence of spontaneous apneas and hypopneas [14, 17]. However, these mice also suffer from comorbidities like arterial hypertension, hyperinsulinemia and hypercholesterolemia, which makes it difficult to ascribe observations to airway obstruction [15].

Here we test the hypothesis that bulking agent injection into the tongue of lean C57BL/6 mice would result in the development of features of OSA, such as intermittent inspiratory flow limitation (IFL), apneas, and hypoxia. We also tested if the severity of these breathing disorders may correlate with the severity of systolic and diastolic dysfunction and cardiac CaMKII expression in these lean mice devoid of comorbidities.

## Materials and methods

All detailed method protocols and data are available upon reasonable request by the corresponding author. All investigations are conformed to directive 2010/63/EU of the European Parliament. The investigation conforms to the Guide for the Care and Use of Laboratory Animals published by the US National Institutes of Health (NIH Publication No. 85–23, revised 1985) and to local institutional guidelines. The protocol was approved by the Committee on the Ethics of the government of Unterfranken, Bavaria, Germany (Protocol Number: 55.2-2532-2-512). All animals were euthanized by cervical dislocation during the light period, i.e. regular sleep time of the animals (usually in the afternoon).

### Polytetrafluoroethylene injection into the tongue

Fifty-nine male C57BL/6 mice at an age of 8–12 weeks (Charles River Laboratories) were included into the study. 31 mice were allocated to polytetrafluoroethylene (PTFE; 35 μm particle size; Sigma Aldrich) injection into the tongue and 28 to control (no treatment, S1 Fig). PTFE has been frequently used as an inert bulking agent for the treatment of primary vesicoureteral reflux in patients [18].

Our approach was based on the findings of Brennick et al., who had measured pharyngeal structures of New Zealand Obese mice (NZO) with spontaneous OSA using MRI [14]. Interestingly, they report that NZO mice (aged 23 weeks, mean body weight 35.7 g) showed a significantly increased mean tongue volume to about 137 μl (compared to 104 μl in control animals). This corresponds to a mean increase of 33 μl tongue volume. Since the tongue volume is the most important determinant of pharyngeal airway size for OSA [19], we aimed to increase the tongue volume of our mice to a similar extent by PTFE injection into the base of the tongue. We used younger mice (mean body weight 27.7 g, only about 70% of the body weight compared to Brennick et al. [14]) to enable the 8-week follow-up observation period. Thus, we anticipated that an increase of about 20–25 μl tongue volume would result in a similar airway obstruction. PTFE is a solid substance (density 2.1 g/ml). 50 mg of PTFE was diluted to 100 μl (50% w/v) with glycerol (Sigma Aldrich). 100 μl of this dilution contains 24 μl pure PTFE, which almost exactly matches the aimed increase in tongue volume. Larger injection volumes were investigated in some test mice, but periprocedural mortality exceeded. Since we were not interested in less upper airway obstruction, we have not studied lower injection volumes.

The allocation to the two treatment groups was done on a random basis. For each day of surgery, a similar number of mice was allocated to both groups. Mice were treated 1 h before the PTFE injection with buprenorphine (0.1 mg/kg bodyweight (BW) intraperitoneal) for optimal analgesia. Mice were anesthetized using intraperitoneal injections of medetomidine (0.5 mg/kg), midazolam (5 mg/kg) and fentanyl 0.05 mg/kg BW. After establishment of anesthesia, the mice were placed in a supine position on a heating plate and body temperature was controlled by a rectal probe. In addition, anesthesia was controlled by continuous monitoring of respiration and ECG. The mouse tongue was manually pulled as far as possible out of the oral cavity to allow for access to the base of the tongue. By using a 27-gauge cannula, about 100 μl PTFE dilution were then injected in the base of the tongue. In order to increase the tongue diameter as balanced as possible, the total volume of 100 μl was divided into multiple injections into depots at the dorsal and ventral side of the tongue (S1 Video). At the end of the procedure, anesthesia was antagonized using intraperitoneal injections of atipamezole (2.5 mg/kg), flumazenil (0.5 mg/kg) and buprenorphine (0.1 mg/kg BW). Untreated littermates were used as control mice.

From 31 mice treated with PTFE, 6 mice had to be killed within 72 h because of surgery-related complications (e.g. bleeding into the tongue, extensive tongue enlargement or infection). In order to respect animals' wellbeing and to avoid animal suffering, we performed everyday visual inspection of every mouse. In particular, we analyzed their skin, food intake, movements and interaction with other mice. If a mouse showed an abnormal behavior, we immediately sacrificed the animal (6 mice had to be sacrificed (S1 Fig)). All the other mice (25/31) showed no evidence of stress or pain and could be monitored for the whole observation period of 8 weeks.

## Sonographic measurement of tongue diameter

Tongue size was measured by ultrasound during the PTFE injection procedure. Mice were placed in supine position onto a heating plate. The tongue was gripped with a tiny crocodile clip. Ultrasound gel was placed onto the murine throat, mandible and mouth, but not on nostrils to keep mice breathing. Thereafter, a 30 MHz center frequency transducer (Vevo3100 system from VisualSonics, Toronto, Canada) was placed at median position of the murine throat to measure the dorso-ventral tongue diameter in sagittal plane. For some recordings, the ultrasound head was rotated clockwise by 90˚ to also measure the lateral tongue diameters in the

transversal plane (S2A Fig). Recordings were acquired with 56 frames/s (gain 30 dB). For optimal magnification, acquisition was performed with 10.00 mm depth and 15.36 mm width. We used the presetting of VisualSonics; thus, no calibration was required. By carefully stirring the tongue via the crocodile clip and comparing tongue movements with the other pharyngeal structures under sonographic recording, tongue surface was easily discriminated from surrounding tissue and tongue diameter was assessed. Similar measurements were done before and after PTFE injection in a standardized manner. All measurements were done by the same investigator; therefore, Kappa statistics cannot be reported. We did not use any fluorescence techniques to identify the area of injection.

## Monitoring of spontaneous breathing

Two weeks after the PTFE injection, spontaneous breathing was recorded by whole-body plethysmography (Buxco Electronics, Harvard Bioscience, Holliston, MA, USA) and analysis of the continuous box flow (FinePointe software, version 2.4.6.9414). To test whether breathing parameters remain stable for the whole observation period, whole-body plethysmography was repeated for some mice 8 weeks after PTFE injection. Mice were placed in the whole-body plethysmography chamber (9 cm diameter and 8 cm height), which was designed with one whole-body plethysmography port, one port for a drinking bottle and one port for an air exhauster (0.2 liter per minute; Buxco bias flow regulator). The system was calibrated according to the manufactures guidelines. Since mice are nocturnal animals, continuous recordings (sampling frequency 1 kHz) were done for 8 h during day-time, the interval with the highest frequency and duration of sleep periods complying with the murine sleep cycle [20]. This constitutes an established alternative method for sleep apnea monitoring if polygraphy with electroencephalography is not possible [20]. In addition, a few mice were subjected to whole-body plethysmography during night-time to compare frequencies of apneas and inspiratory flow limitations when mice are awake.

For a subgroup of mice, inspiratory flow limitations (IFLs) were analyzed using FinePointe software as following: for each breath, we calculated the ratio of inspiration time and tidal volume. A breath was considered to be flow limited if this ratio was increased at least 2.576 standard deviations compared to a running mean of the last 100 breaths, which corresponds to the 99% confidence interval. This corresponds to either an increase in inspiration time or a decrease in tidal volume or a combination of both. The cut-off was chosen after comprehensive manual investigation of box flow recordings. In addition, to avoid false positive detections, the flow was only considered to be flow limited if its peak inspiratory flow was at least 2.576 standard deviations lower than the mean peak inspiratory flow of the last 100 breaths. The absolute frequency (/h) and the proportional frequency (%) of IFLs were calculated as number of IFLs normalized to either total observation period or total number of breaths, respectively.

We further calculated the number of IFL aggregates that were defined by 3 or more consecutive IFLs and reported their absolute frequency (/h).

For apnea detection, an automatic detection algorithm (apnea analysis module of FinePointe) was used. We defined an apnea as a cessation of breathing for at least 1 s [21]. The frequency of apneas (/h) was calculated as number of apneas normalized to total observation period. Apnea frequency was considered to be abnormally increased, when being at least 2 standard deviations greater than the mean apnea frequency in control mice.

## Isolation of RNA and transcription into cDNA

RNA was isolated from ventricular myocardium using the RNeasy Mini Kit (Qiagen, catalog number 74106) like described in the manufacture's manual. RNA was measured using

spectrophotometry (A = 260 nm, NanoDrop™ 2000c, Thermo Scientific™) and cDNA was obtained from 1 μg RNA using random primers (Promega, catalog number C1181), PCR nucleotide mix (Promega, catalog number C1145), RNasin® ribonuclease inhibitor (Promega, catalog number N2115), reverse transcriptase (Promega, catalog number M170B), and reverse transcriptase 5x reaction buffer (Promega, catalog number M531A). The ingredients were incubated for 1 h at 37° C according to the manufactures' guidelines.

## Quantification of KDM6A and HIF1α

KDM6A and HIF1α mRNA expression were measured in ventricular myocardium of PTFE and control mice by real-time qPCR with cDNA (see above) on ViiA 7 real-time PCR system (Applied Biosystems). As described by Jung et al., HIF1α mRNA expression increases under hypoxic conditions in cardiomyocytes [22]. In contrast to HIF1α protein analysis, mRNA expression is less vulnerable to fluctuation due to the critical timing and method of the euthanasia, which is why this substrate was selected for analysis. Experiments were performed with TaqMan™ Fast Advanced Master Mix (Applied Biosystems) and the following settings were used accordingly to the manufacture's manual: initial uracil-N-glycosylase incubation at 50° C (2 min), polymerase activation at 95° C (2 min), followed by 40 cycles with 95° C (1 s) and 60° C (20 s). Pre-designed TaqMan® Gene Expression Assays (Applied Biosystems) were used for quantification of HIF1α (assay ID Mm00468869_m1), KDM6A (assay ID Mm00801998_m1) and β-actin (assay ID Mm01205647_g1).

All samples were measured as triplicate and the average threshold cycle (Ct) was used for the comparative Ct relative quantification analysis method [23]. Therefore, the mean Ct of each target was subtracted from the corresponding mean Ct of the housekeeper β-actin, obtaining the delta Ct (dCt) value. Calculating $2^{-dCt}$ x 100 revealed the relative expression of each target (in % of β-actin).

## Transthoracic echocardiography

Transthoracic echocardiography was performed blinded using a Vevo3100 (VisualSonics, Toronto, Canada) system with a 30 MHz center frequency transducer. The animals were initially anesthetized with 2% isoflurane (Isoflurane Vaporizer; VisualSonics, Toronto, Canada), while temperature-, respiration-, and ECG-controlled anesthesia was maintained with 1.5% isoflurane. Two-dimensional cine loops with frame rates of >200 frames/s of a long axis view and a short axis view at mid-level of the papillary muscles as well as M-mode loops of the short axis view were recorded. Left ventricular ejection fraction (parasternal long axis view M-mode), as well as peak early (E) and late (A) diastolic filling velocities (PW doppler) were measured. Tissue doppler mode was used to measure peak early diastolic (e') and late diastolic (a') mitral annular velocities. The ratio E/e' was calculated to estimate the severity of diastolic dysfunction. Measurements were obtained by an examiner blinded to the treatment of the animals.

## Western blots

Tris buffer was used to homogenize the whole left and right murine ventricle. The buffer contained (mmol/L) 20 Tris-HCl, 200 NaCl, 20 NaF, 8.9 Nonidet P-40 (Sigma Aldrich), 18.3 phenylmethanesulfonylfluoride (Sigma Aldrich), complete protease inhibitor cocktail (Roche) and complete phosphatase inhibitor cocktail (Roche). Protein concentration was measured by BCA assay (Pierce Biotechnology). Proteins were denatured at 95° C for 5 min (500 rpm) in 2% β-mercaptoethanol (Sigma Aldrich). They were then separated on 8% SDS-polyacrylamide gels, transferred to a nitrocellulose membrane (GE Healthcare) and

incubated overnight with the primary antibody at 4˚ C: mouse monoclonal anti-CaMKII (1:1000, BD Biosciences, catalog number 611293) and anti-GAPDH (1:20000, Abcam, catalog number G8795). After that, the samples were incubated for 1 h at room temperature with the secondary antibody (HRP-conjugated sheep anti-mouse IgG: 1:3000 for anti-CaMKII, 1:30000 for anti-GAPDH, GE Healthcare, catalog number NA931VS). After incubation with Immobilon™ Western Chemiluminescent HRP Substrate (Millipore) for 5 min at room temperature, protein bands were developed onto Super XR-N X-ray films (Fujifilm) and scanned by ChemiDoc™ MP Imaging System (Bio-Rad). Mean densiometric values were determined using ImageJ.

### Data analysis and statistics

All measurements and experiments were performed and analyzed blinded to the treatment group (control or PTFE) and to frequency of apneas. Experimental data are presented as means ± standard error of the mean (SEM). All statistical analyses were based on the number of mice and normal distribution was assessed by Shapiro-Wilk normality test. Parametric or non-parametric tests were applied to test for significant differences, depending on whether a variable was normally distributed or not. Parametric and non-parametric tests used for the comparison of two groups were Student's t and Mann-Whitney test, respectively. Ordinary one-way ANOVA with Holm-Sidak's post-hoc correction and Kruskal-Wallis test with Dunn's post-hoc correction were used for comparisons of more than two groups that were either normally or not normally distributed, respectively. One-way repeated measures ANOVA with Holm-Sidak's post-hoc correction was used for the comparison of paired data that was normally distributed. If more than two groups and two different factors were compared in a repeated measures design, mixed-effects model analysis with Holm-Sidak's post-hoc correction was used. Chi-square test was used for the comparison of categorial data. The tests above as well as linear regression analyses were used in GraphPad Prism 8 to test for significance, as appropriate. Two-sided P-values below 0.05 were considered as statistically significant.

## Results

### Increased tongue diameter after PTFE injection

Injection of polytetrafluoroethylene (PTFE) into the tongue of lean male C57BL/6 mice at an age of 8–12 weeks led to a sustained and significant increase of the sagittal tongue diameter from 2.75±0.16 mm to 3.67±0.20 mm (N = 31; P = 0.002; Fig 1A and 1B). Interestingly, we observed a similar increase in transversal tongue diameter leading to a homogenous increase of cross-sectional tongue area from (in mm$^2$) 9.23±0.41 to 19.90±0.86 (N = 5; P<0.001; S2A Fig). To investigate whether the PTFE depots remain in the tongue for the whole follow up period, we measured the tongue diameter again in a subgroup of mice at 8 weeks after injection. Interestingly, the tongue diameter was still significantly thicker (3.58±0.18 mm; N = 10; P = 0.04).

An increased tongue diameter may disturb food intake. Therefore, we monitored the body weight (BW) of a subset of mice for 8 weeks. At 8 weeks follow up, we observed a similar significant increase in BW in control (from 27.86±0.54 g to 31.29±0.64 g; N = 19; P<0.001; Fig 1C) and PTFE injected mice (from 27.72±0.61 g to 30.85±0.59 g; N = 24; P<0.001). Importantly, mixed-effects model analysis accounting for timepoint and intervention group revealed no difference in BW between PTFE and control mice, neither at baseline nor at 8 weeks follow up (P (time)<0.001; P(intervention) = 0.82; P(interaction) = 0.89; Fig 1C).

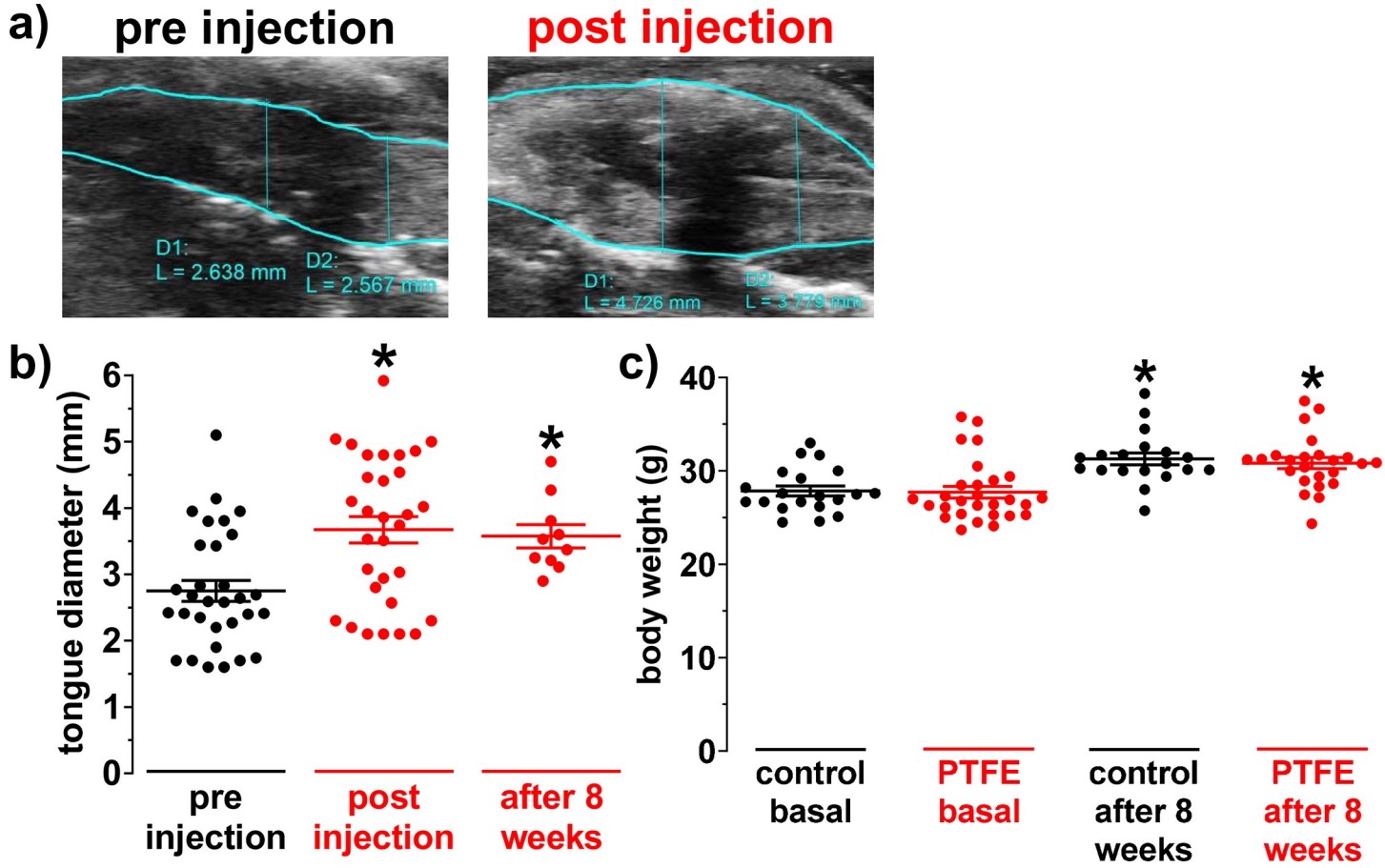

**Fig 1. Increased tongue diameter after PTFE injection.** a) Original ultrasound image of a murine tongue at median position in sagittal plane before and after polytetrafluoroethylene (PTFE) injection. The green line indicates the contour of the tongue. b) Mean dorso-ventral tongue diameter of 31 mice before and after PTFE injection. Interestingly, the mean dorso-ventral tongue diameter remained significantly increased at 8 weeks after the injection (N = 10). c) Due to a random allocation, there was no difference in body weight between control (N = 20) and PTFE mice (N = 28) at baseline. Importantly, there was a similar and significant increase in body weight in both control (N = 19) and PTFE mice (N = 24) at 8 weeks after the injection. *—P<0.05 vs. pre injection (b) or corresponding basal value (c), Kruskal-Wallis test with Dunn's post-hoc correction (b) and mixed-effects model analysis with Holm-Sidak's post-hoc correction (c).

### PTFE injection increases the frequency of inspiratory flow limitations and apneas

Two weeks after the PTFE injection, mice were subjected to whole-body plethysmography for apnea analyses (Fig 2A). Interestingly, sleeping PTFE mice (during day-time) exhibited a significantly increased frequency of apneas of 11.32±1.45 per h (N = 25), compared to their littermates with 6.27±0.80 apneas per h (N = 28; P = 0.004; Fig 2B). Moreover, the proportion of mice showing an abnormally increased apnea frequency above the cut-off of 14.75 apneas/h (mean apnea frequency of control mice + 2 standard deviations) was significantly increased in PTFE-injected mice (S2B Fig). Interestingly, 8 out of 25 PTFE mice but only 2 out of 28 control mice showed an abnormally increased apnea frequency (P = 0.02; S2B Fig). In order to further validate our novel method and the mechanism of the increase in apnea frequency, we conducted a linear regression analysis with the tongue diameter after PTFE injection and the corresponding apnea frequency (Fig 2C). Intriguingly, there was a significant positive correlation between those parameters ($R^2$ = 0.22; N = 25; P = 0.02). In addition, we investigated inspiratory flow limitations (IFLs) as a milder form of upper airway obstruction (Fig 2D and S3 Fig).

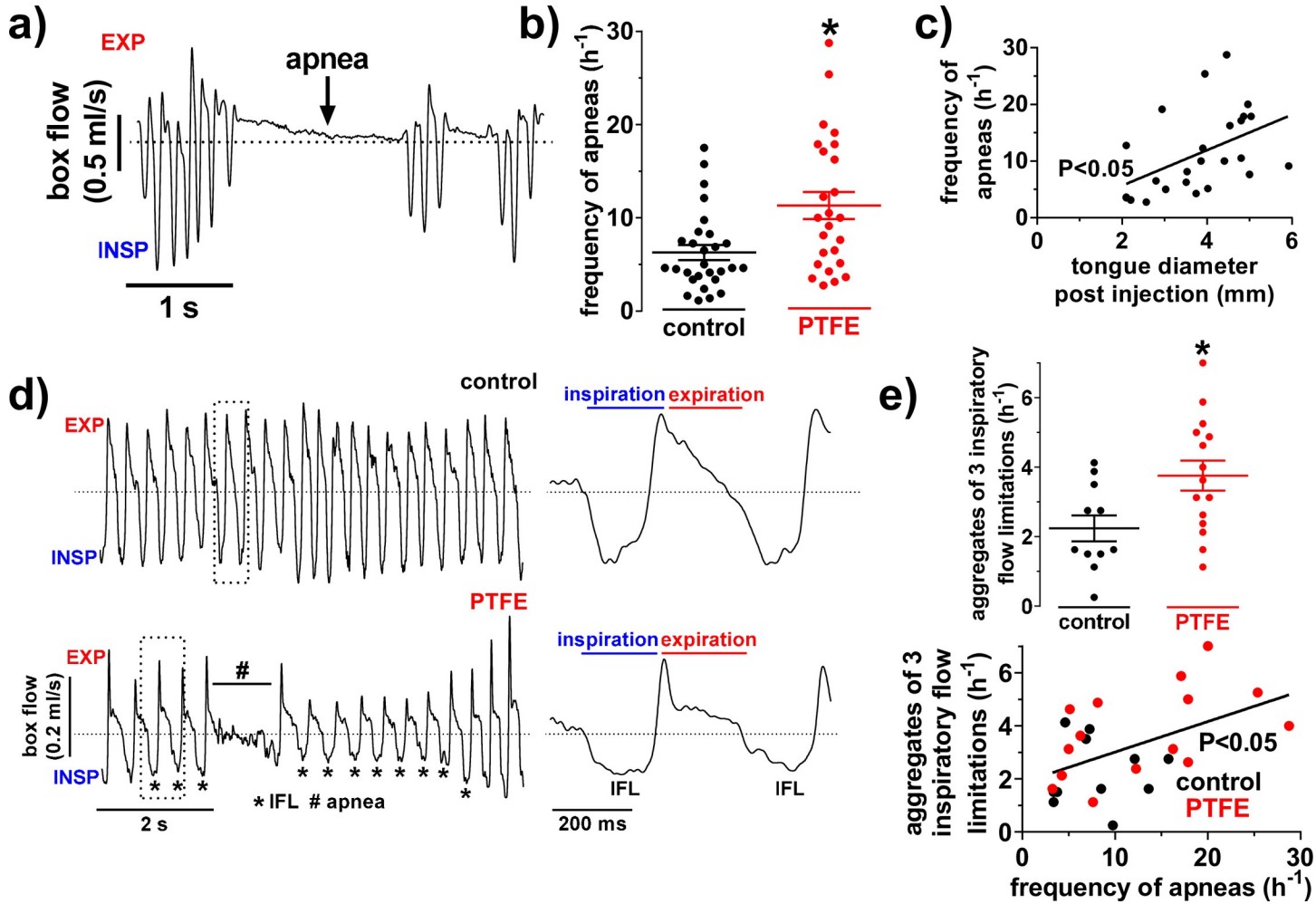

**Fig 2. PTFE injection into the tongue induces apneas and inspiratory flow limitations.** a) Representative original recording and (b) mean data of apnea frequency. Interestingly, polytetrafluoroethylene (PTFE) injected mice (N = 25) had a significantly increased frequency of apneas compared to control (N = 28). c) Linear regression analysis showing a positive correlation between the tongue diameter after PTFE injection and the frequency of apneas (N = 25), underlining the obstructive character of the observed apneas. d) Representative original breath recordings of a control (upper panel) and a PTFE-treated mouse (lower panel). The latter developed multiple breaths with inspiratory flow limitation (IFL), which is a typical sign of airway obstruction. e) Interestingly, we observed a significantly increased frequency of IFL aggregates in PTFE-treated mice, leading to a significant positive correlation with the frequency of apneas (N = 26). *—P<0.05, Mann-Whitney test (b), Student's t-test (e), and linear regression analysis (c+e).

When mice were awake (during night-time, 10 p.m.– 7 a.m.), control and PTFE mice showed a similar breathing pattern with a negligible number of IFLs and no IFL aggregates indicating that PTFE injection into the tongue does not induce a fixed upper airway obstruction (S4 Fig). In conscious mice, mean IFL frequency (/h) was 1.98±0.52 vs. 2.49±0.51 (PTFE vs. control; N = 5 for both; P = 1.00; in S4C Fig). Also, mean apnea frequency (/h) was very low and similar in awake PTFE and control mice (1.95±1.01 vs. 2.38±1.19; N = 5 for both; P = 1.00; S4C Fig). In contrast, when analyzing sleeping mice (during day-time), PTFE injection resulted in a significant increase in IFL frequency (/h) from 57.74±4.58 to 72.28±3.59 (P = 0.02) and IFL aggregate frequency from 2.24±0.37 to 3.76±0.43 (N = 11 vs. 15; P = 0.02; S3A Fig and Fig 2E). Moreover, the apnea frequency correlated significantly positive with the IFL aggregate frequency in sleeping mice ($R^2$ = 0.24; N = 26; P = 0.01; Fig 2E), suggesting that these apneas may be due to intermittent airway obstruction. The latter is further supported by the clustered occurrence of IFLs and apneas that can be found only in the murine sleeping period (S4 Fig). In

accordance, the percentage of flow limited breaths was significantly increased in sleeping mice from 0.43±0.028 to 0.51±0.02 (PTFE vs. control, P = 0.04), leading to a significant positive correlation with the frequency of apneas ($R^2$ = 0.20; P = 0.02; S3B Fig). Importantly, the intermittent airway obstruction in sleeping mice remained stable for the whole observation period. Compared to the 2-week timepoint, frequencies of apneas (P = 0.71), IFLs (P = 0.38), and IFL aggregates (P = 0.95) were similar at 8 weeks after PTFE injection (N = 6 for all; S2C Fig).

## PTFE injection resulted in decreased KDM6A and increased HIF1α expression

Lysine (K)-specific demethylase 6A (KDM6A) and hypoxia-inducible factor 1α (HIF1α) have been shown to be valuable markers for hypoxemia [22, 24–27]. We measured cardiac KDM6A and HIF1α mRNA expression by qPCR. KDM6A has been shown to be negatively correlated with the extent of hypoxemia [27]. We observed a significant reduction in the expression of KDM6A (in % relative to β-actin) in PTFE mice from 2.44±0.46 (N = 5) to 1.28±0.19 (N = 8; P = 0.02; Fig 3A), leading to a significantly negative correlation with the frequency of apneas ($R^2$ = 0.61; N = 13; P = 0.002; Fig 3A). This indicates that the observed apneas may be responsible to induce hypoxemia. Consistent observations were made with the established hypoxemia marker HIF1α. Compared to control, there was a significant increase in HIF1α mRNA (in % relative to β-actin) in PTFE mice from 6.20±1.01 (N = 5) to 16.47±1.95 (N = 8; P = 0.002; S5A Fig). Moreover, we found that the frequency of apneas strongly and significantly correlated with HIF1α mRNA expression ($R^2$ = 0.68; N = 13; P<0.001; S5A Fig). Interestingly, the tongue diameter also corelated significantly negative with KDM6A ($R^2$ = 0.35.; N = 13; P = 0.03; Fig 3B) and significantly positive with the HIF1α expression ($R^2$ = 0.62; N = 13; P = 0.001; S5B Fig), supporting our novel methodological approach of tongue enlargement with PTFE.

## Systolic and diastolic dysfunction in PTFE-treated mice

Since OSA may impair systolic and diastolic contractile function [2, 28], we assessed cardiac function by echocardiography in mice at 8 weeks follow up (Fig 4 and Table 1). PTFE injected

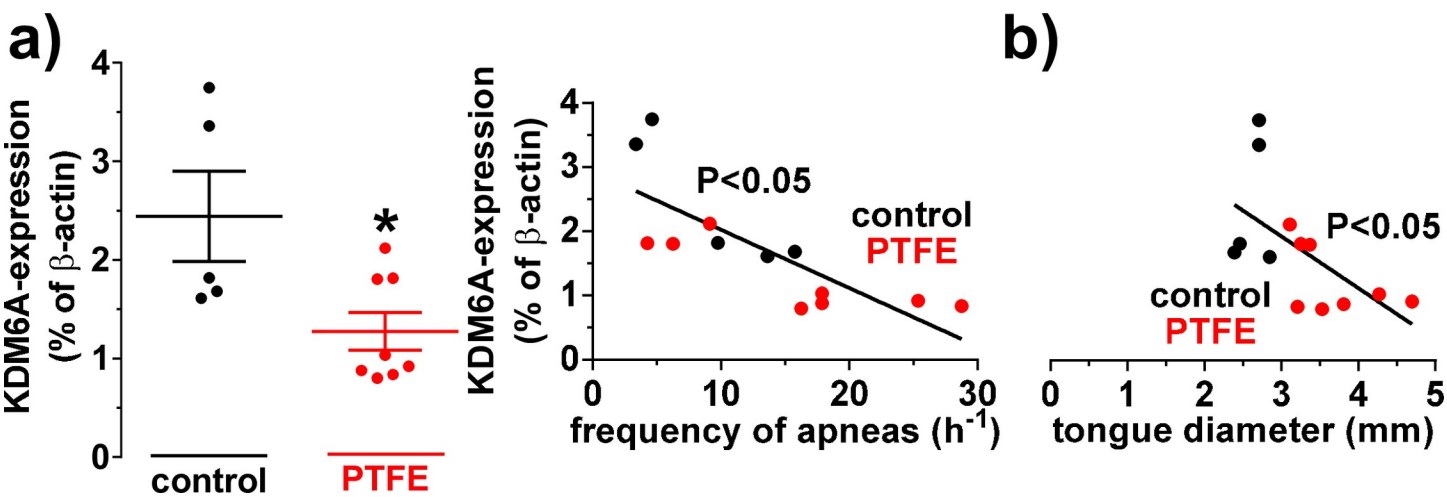

**Fig 3. KDM6A mRNA expression is decreased in PTFE mice.** KDM6A mRNA expression was analyzed by qPCR (normalized to β-actin) from hearts. a) Scatter plots of KDM6A mRNA expression in control (N = 5) and PTFE-treated (N = 8) animals (left panel). There was a significant downregulation of KDM6A mRNA expression after PTFE treatment. The level of KDM6A mRNA expression correlated significantly negative with the frequency of apneas, suggesting hypoxemia (right panel). b) Interestingly, the tongue diameter correlated significantly negative with the KDM6A expression, indicating upper airway obstruction to potentially induce apnea-dependent hypoxemia. * —P<0.05, Student's t-test and linear regression analysis, as appropriate.

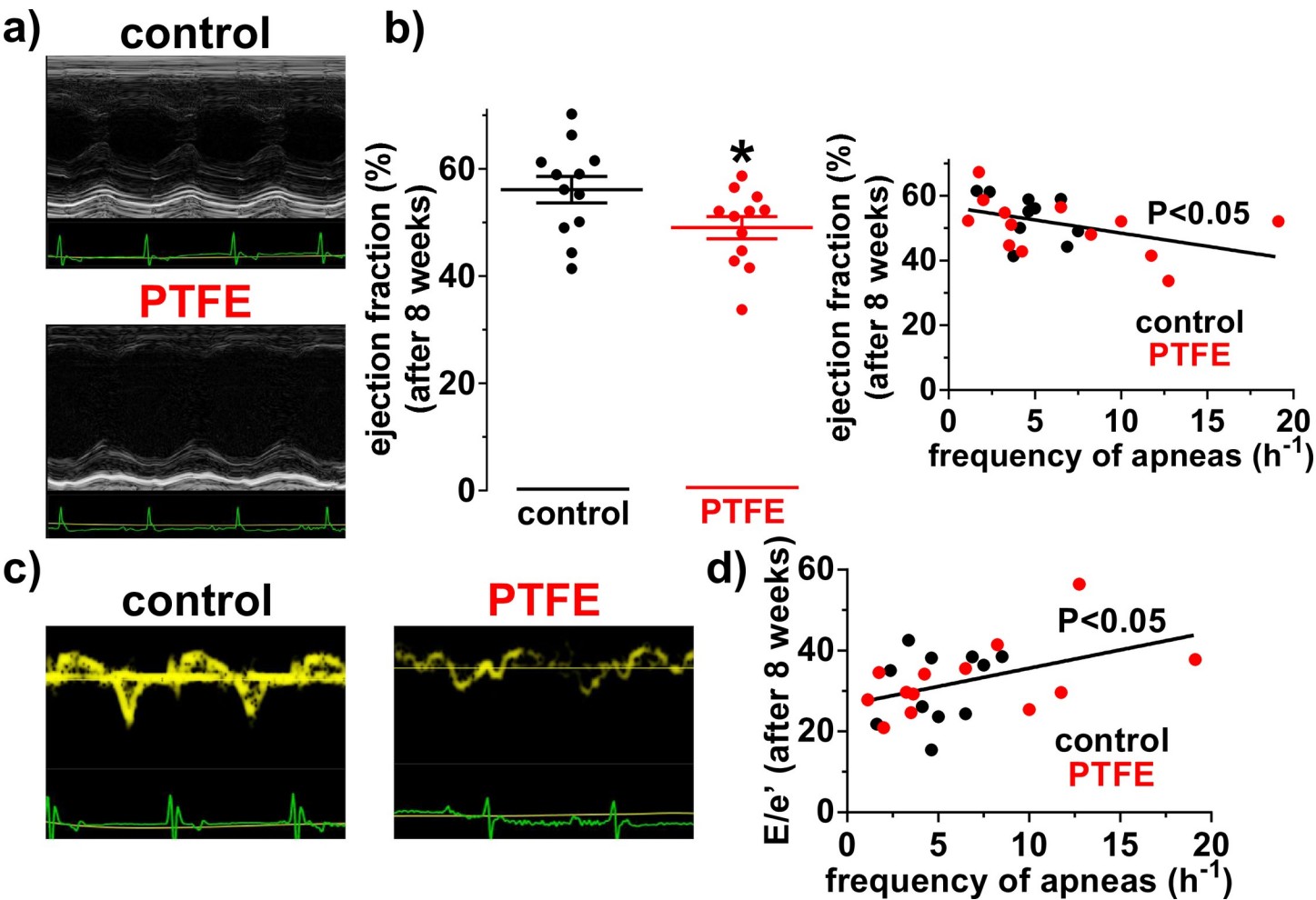

**Fig 4. Systolic and diastolic dysfunction in PTFE mice.** a) Representative M-mode echocardiographic recordings and (b) mean data for ejection fraction, which was significantly reduced in polytetrafluoroethylene (PTFE) mice (N = 12 vs. 12; left panel). Furthermore, there was a significant negative correlation between the frequency of apneas and the ejection fraction (right panel). c) Representative recordings of the mitral annular ring velocity with a tissue doppler. d) Interestingly, the frequency of apneas correlated significantly with the ratio E/e' (N = 24), suggesting a more pronounced diastolic dysfunction in mice with more frequent apneas. *—P<0.05, Student's t-test and linear regression analysis, as appropriate.

**Table 1. Echocardiographic parameters.**

|  | Control | PTFE | P Value |
|---|---|---|---|
|  | (N = 12) | (N = 12) |  |
| Heart rate (/min), mean±SEM | 490.75±15.02 | 489.42±12.29 | 0.95[T] |
| Cardiac output (ml/min), mean±SEM | 18.47±1.15 | 19.16±0.89 | 0.64[T] |
| Stroke volume (μl), mean±SEM | 37.47±2.11 | 38.52±1.93 | 0.93[MW] |
| LV end-diastolic diameter (mm), mean±SEM | 3.98±0.07 | 4.22±0.09 | 0.06[T] |
| LV end-diastolic volume (μl), mean±SEM | 68.32±3.22 | 81.42±3.97 | **0.02[T]** |
| Diastolic anterior wall thickness (mm), mean±SEM | 0.87±0.08 | 0.81±0.05 | 0.50[T] |
| Diastolic posterior wall thickness (mm), mean±SEM | 0.68±0.07 | 0.65±0.03 | 0.78[T] |

LV–left ventricular, MW–Mann-Whitney test, T–Student's t-test.

mice showed a significantly decreased left ventricular ejection fraction of 49.02±2.07% (N = 12) compared to control mice with 56.10±2.49% (N = 12; P = 0.04; Fig 4B). In accordance, left ventricular end-diastolic volume was significantly increased in PTFE mice (Table 1). The ejection fraction also correlated significantly negative with the frequency of apneas ($R^2$ = 0.19; P = 0.04; Fig 4B).

Furthermore, we analyzed the ratio of the early diastolic filling velocity (E) and the peak early diastolic mitral annular velocity (e') to estimate the severity of diastolic dysfunction. Interestingly, E/e' correlated significantly positive with the frequency of apneas ($R^2$ = 0.19; N = 24; P = 0.04; Fig 4D).

## Increased heart and lung weight in PTFE-treated mice

Contractile dysfunction is frequently accompanied by structural changes of the heart and clinical signs of heart failure. Therefore, we measured heart (HWs) and lung weights (LWs) and normalized it to the BW (Fig 5) at 8 weeks follow up that can be used to estimate cardiac hypertrophy and pulmonary edema, respectively [29, 30]. Interestingly, HW/BW (Fig 5A) was not only significantly increased from 0.59±0.03% (N = 19) to 0.68±0.03% in PTFE injected mice (N = 24; P = 0.03), but also correlated significantly positive with the frequency of apneas ($R^2$ = 0.10; N = 39; P = 0.045). Importantly, this increase in heart weight appears to be rather due to eccentric hypertrophy with enlarged left ventricular end-diastolic volume and not due to hypertrophic wall thickening (Table 1).

In accordance with reduced left ventricular contractile function, PTFE-treated mice also showed a significant increase in LW/BW (Fig 5B) consistent with lung edema. Compared to control, LW/BW was significantly increased in PTFE-treated mice (0.64±0.02%; N = 24 vs. 0.56±0.02%; N = 19; P<0.001). In addition, LW/BW correlated significantly positive with the frequency of apneas ($R^2$ = 0.13; N = 39; P = 0.02).

## Increased CaMKII expression in PTFE-treated mice

Increased CaMKII expression has been shown in multiple studies to be implemented in the pathogenesis of heart failure, hypertrophy, and arrhythmias [8, 9, 31], and has also been found in patients with sleep-disordered breathing [4]. Therefore, we analyzed CaMKII expression in ventricular homogenates of PTFE-treated (N = 6) and control mice (N = 9) at 8-week follow up by Western blotting (Fig 6). Interestingly, compared to control, PTFE-treated mice showed a significant increase in CaMKII expression normalized to GAPDH (5.01±0.61 vs. 2.51±0.66, P = 0.02; Fig 6B). Furthermore, the CaMKII/GAPDH expression correlated significantly positive with the frequency of apneas ($R^2$ = 0.28; P = 0.04; Fig 6C).

## Discussion

In this study, we describe a novel mouse model of OSA by injecting polytetrafluoroethylene (PTFE) into the tongue of lean male C57BL/6 mice. These mice develop IFLs with an increased frequency of apneas. At 8 weeks follow up, PTFE-treated mice show increased ventricular expression of the novel hypoxia marker KDM6A, mild systolic and diastolic contractile dysfunction, signs of cardiac eccentric hypertrophy and pulmonary edema, that were accompanied by CaMKII overexpression. In the absence of co-morbidities, this mouse model may be suitable to investigate the mechanisms of spontaneous obstructive sleep apnea.

OSA is a widespread disease with increasing prevalence [1]. Since it is frequently associated with multiple disorders like heart failure or arrhythmias leading to increased morbidity and mortality, it takes on even greater socio-economic significance [2, 3]. To date, the treatment of OSA is mainly limited to continuous positive airway pressure (CPAP), but acceptance of

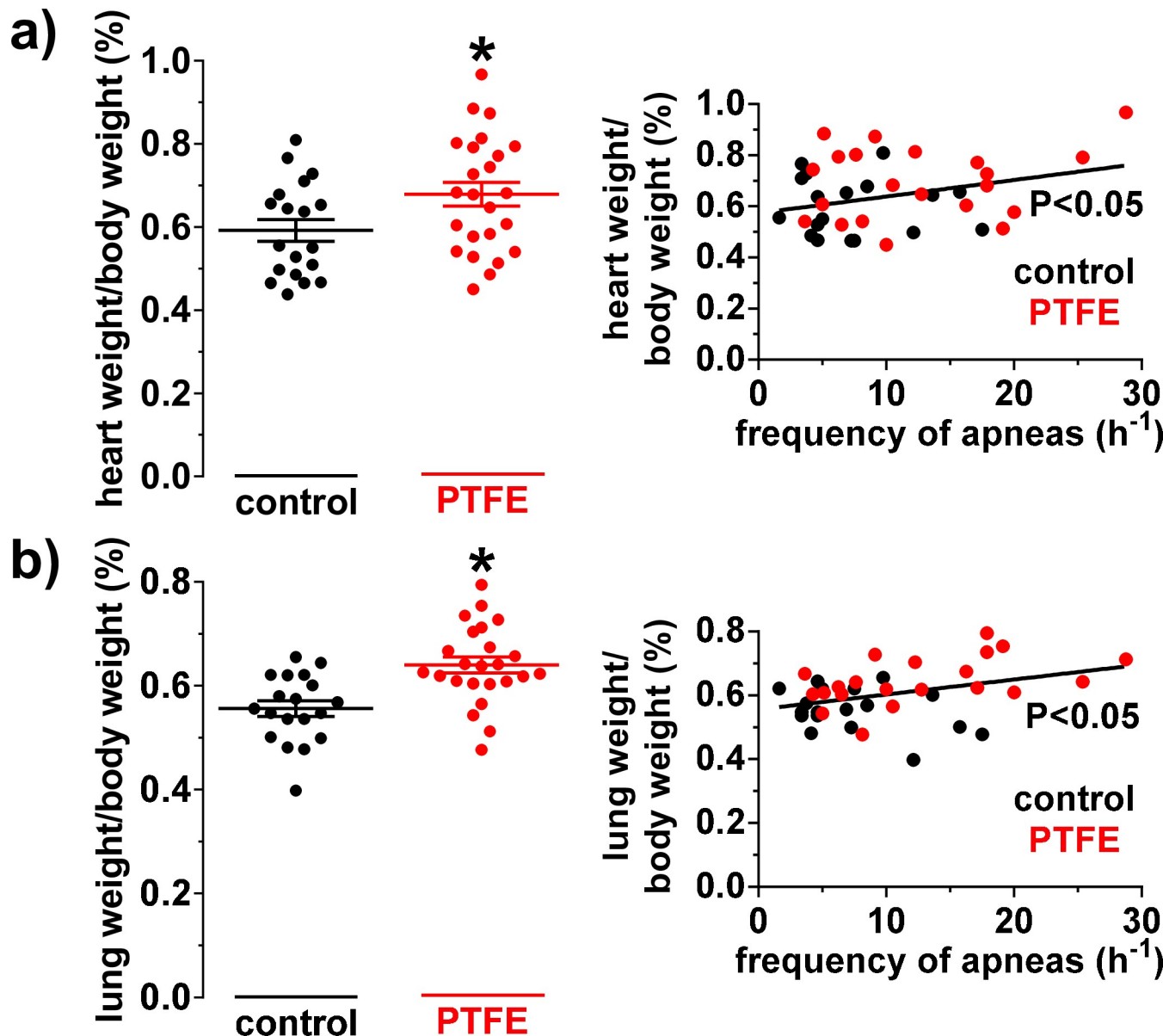

**Fig 5. Heart and lung weight in control and PTFE injected mice.** a) Mean heart weight/body weight ratio (in %) in PTFE (N = 24) and control mice (N = 19). Interestingly, the heart weight was significantly increased in PTFE mice (left panel) and correlated significantly with the frequency of apneas (N = 39; right panel). b) Mean lung weight/body weight ratio (in %) is shown. Intriguingly, the lungs were not only significantly heavier in PTFE mice (N = 24 vs. 19; left panel), but corelated also positively with the frequency of apneas (N = 39; right panel). *—P<0.05, Student's t-test and linear regression analysis, as appropriate.

CPAP in patients with low symptom burden is limited [6] and treatment with positive pressure ventilation (adaptive servo-ventilation) may be even harmful for specific patient populations (e.g. for patients with predominant central apneas) [7]. It is consequently essential to find novel therapeutic concepts for the treatment of OSA and OSA-related diseases. Therefore, it is necessary to get a better understanding about the mechanisms of OSA affecting the cardiovascular system. Although there are multiple studies analyzing the mechanisms OSA in humans, they always struggle with the problem of patient heterogenicity and several comorbidities as

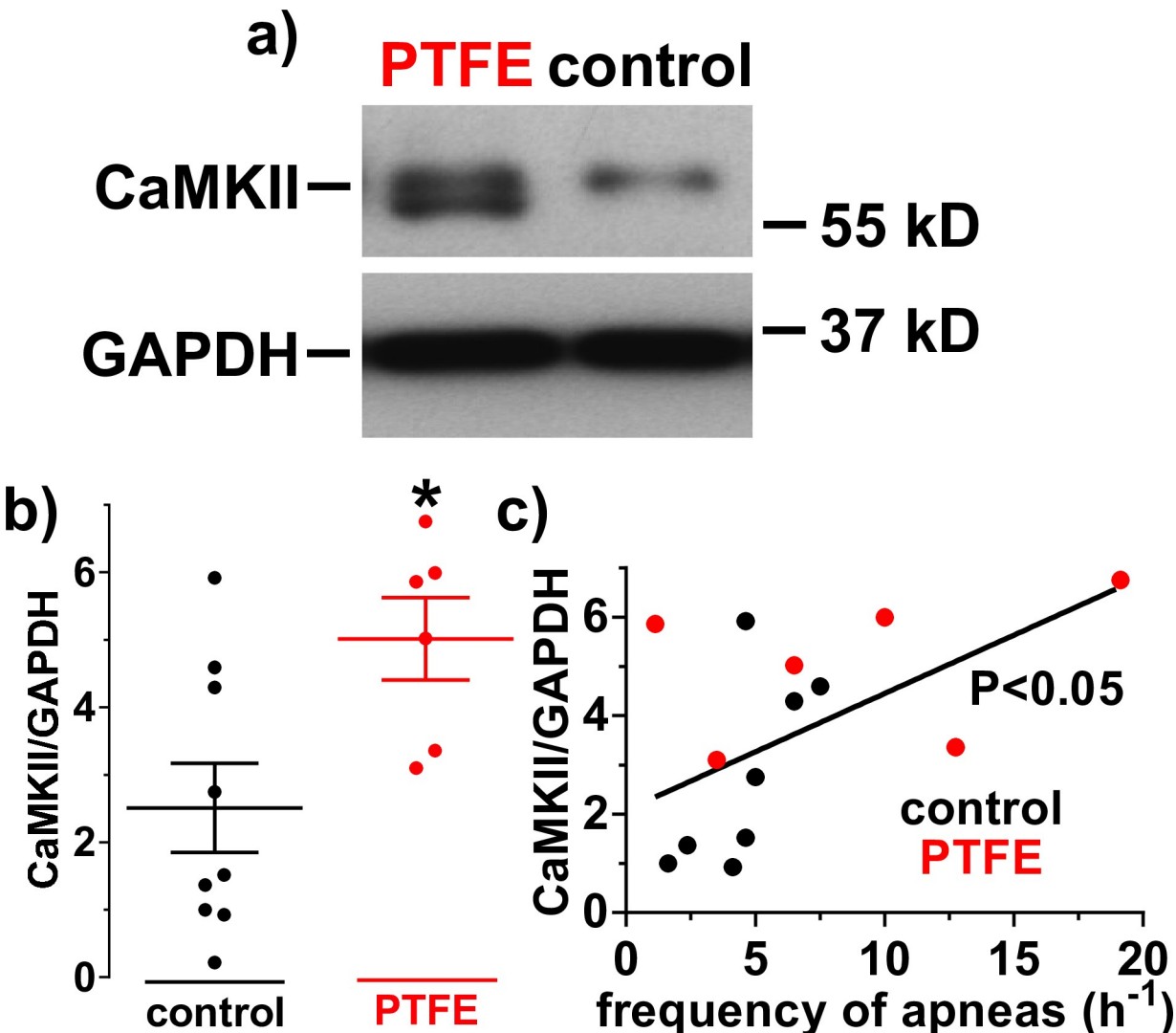

**Fig 6. Increased CaMKII expression in PTFE mice.** a) Original Western blots investigating Ca/calmodulin-dependent protein kinase II (CaMKII) expression in ventricular homogenates. b) Mean densitometric values show a significantly increased CaMKII expression in polytetrafluoroethylene (PTFE) injected mice (N = 6 vs. 9). c) Interestingly, there was also a significant positive correlation between the frequency of apneas and CaMKII expression (N = 15). *—P<0.05, Student's t-test (b) and linear regression analysis (c).

potential confounders, which makes it difficult to interpret mechanistic data. To avoid those pitfalls, research of OSA in animal models is important [16].

## Current models of sleep apnea are limited

Despite the urgent need for animal models of sleep apnea, current animal models struggle with many limitations [12–17, 32].

A very common animal model is tracheotomy with intermittent tracheal occlusion, which has already been used in several animals like dogs [33], baboons [34], and rats [16, 20]. Linz et al. have reported tracheal occlusion with a negative pressure of -80 mbar in anesthetized pigs resulting in increased blood pressure and oxidative stress, activation of fibrotic pathways and subsequently an increased burden of atrial fibrillation [12]. However, while this model may be applied in otherwise healthy animals thereby excluding the confounding of

comorbidities, it is currently not available in mice, excluding the opportunity to test novel hypotheses in genetically modified animals. Also, this model requires deep medical sedation and analgesia of the animals, which may interfere with the apnea-dependent sleep fragmentation with sudden awakening (arousal) and consecutive alteration of the autonomous nervous system [3, 16]. Additionally, Crossland et al. have reported a rodent model of OSA by chronic intermittent tracheal occlusion in rats during their sleep cycle (9 AM to 5 PM) [20]. However, all models of tracheal occlusion require substantial material and personal investment that precluded application in large animal cohorts for extended periods of follow up. Another rat model of OSA mimicking airway obstruction was first described by Farré et al. [35]. Rats were placed awake in a setup with two chambers split by a latex neck collar and airway obstruction was induced by interruption of bias flow in the head chamber [35, 36]. While the authors observed remarkable decreases in oxygen saturation, this model is hardly comparable to clinical OSA since rats were placed awake in the chamber [35, 36]. Although this model requires few animal manipulation and the setup is easy to handle, it is very time intensive since animals have to be placed in the obstruction chambers every day for at least three weeks [35, 36].

Chronic intermittent hypoxemia (CIH) is another very frequently used method for investigation of sleep-disordered breathing in animals [16, 37]. Although this method gave valuable insights into the pathophysiology of sleep-disordered breathing, the optimal gas composition is still under discussion [16]. Moreover, it completely lacks important aspects of the pathophysiology like airway obstruction. OSA, is mainly characterized by obstruction of the upper airway with consecutive inefficient breathing effort, intrathoracic pressure swings, which are followed by intermittent hypoxemia and β-adrenergic stress during sudden awakening [3]. This may be one explanation, why some clinical findings of patients with sleep-disordered breathing have not been recapitulated by CIH in animals [3, 16, 37].

For instance, sustained sleep-related arterial hypertension has been shown to occur in dogs only in combination with upper airway obstruction (induced by tracheal occlusion), but not following repetitive arousals or hypoxemia alone [3, 16, 38].

Moreover, short-term CIH was found to even protect the heart by inducing ischemic preconditioning and increasing cardiac contractility [39]. But even after long term CIH exposure, the basal cardiac phenotype may depend on the specific CIH protocol applied and may require additional interventions like ischemia/reperfusion to induce detectable damage [40].

Beside these limitations, CIH also requires substantial material and personal investment (cost-intensive ventilation chamber, huge amount of gas to be exchanged) rendering long term studies for larger animal cohorts very difficult, which limits high throughput animal research [16].

## PTFE tongue injection induces obstructive apneas in mice

Considering the difficulties of current animal models, novel mouse models are clearly warranted. Interestingly, obesity, as one of the most common risk factors for OSA, is accompanied by a narrowed upper airway in humans and several animals [13, 14, 16, 17]. Looking at the structures in the upper airway, Schwab et al. have found that an increased tongue volume is the most important risk structure for OSA and is even independent from sex, age, race, craniofacial size, and parapharyngeal fat [19]. In accordance, New Zealand Obese mice have also an increased tongue volume with consecutive apneas and hypopneas [14, 16, 17]. However, these mice also suffer from OSA-independent diseases like arterial hypertension, hyperinsulinemia and hypercholesterolemia, which makes it difficult to ascribe observations to airway obstruction [15].

Thus, it was a reasonable concept to induce intermittent airway obstruction by increasing the cross-sectional area of the tongue. PTFE is an inert substance that can be used for treating

primary vesicoureteral reflux by narrowing the ureterovesical junction [18]. We show here that a single PTFE injection procedure not only resulted in increased tongue diameters that remain increased across the whole follow up period (Fig 1B) without signs of inflammation or disturbance of food intake, but also that PTFE-treated mice develop spontaneous IFLs and abnormally increased apneas. According to the sustained increase in tongue diameter, we could demonstrate that the increased frequencies of IFLs and apneas remained stable for the whole 8-week observation period (S2C Fig). Importantly, PTFE injection into the tongue does not lead to a fixed upper airway obstruction (S4 Fig). In sleeping mice, only about 0.50% of all breaths in PTFE mice were flow limited (S3B Fig). In accordance with the intermittent nature of obstructive breathing abnormalities, the PTFE-induced IFLs and apneas occurred in clusters and only in sleeping mice (S4A Fig). In contrast, awake PTFE mice exhibit a regular breathing pattern without airway obstruction (S4B Fig). Moreover, we observed no IFL aggregates and similar frequencies of IFLs and apneas after PTFE treatment during awake periods (S4C Fig). Therefore, hypoxia is likely not present in awake mice.

Similar observations were made by Philip et al., who injected liquid collagen into the uvula, tongue, and pharyngeal walls of monkeys leading to more frequent hypopneas during sleep [41]. Since monkeys are not a very common animal model, and are more difficult in housing, this model was not established in OSA research [41]. Additionally, polyacrylamide and sodium hyaluronate injection into the palate of rabbits and rats, respectively, has been shown to induce obstructive sleep apnea by upper airway obstruction [42, 43]. Unfortunately, the availability of appropriate knock-out or transgene models is very limited in both rabbits and rats. In this context, future studies investigating sodium hyaluronate injection into the palate of mice may lead to another promising animal model of OSA.

Therefore, we report here a unique mouse model of OSA that avoids many pitfalls of former models. Our model develops spontaneous apneas and does not require anesthesia or additional interventions. This could be important, when investigating OSA-specific features like sudden awakening with consecutive β-adrenergic stress. Further, the mouse models offer great opportunities by investigating specific knock out or transgene mice.

Importantly, our model is not only very effective in inducing obstructive apneas, but also very efficient since a single intervention resulted in a sustained airway obstruction with IFLs and apneas across the whole observation period of 8 weeks. Thus, large cohorts of animals can be investigated with relatively low material and personal requirements.

## Consequences of PTFE-injection on contractile function

Since OSA is frequently associated to systolic and diastolic dysfunction [2, 28], we performed echocardiography to complete the basal characterization of our new model (Fig 4). PTFE-treated mice show mild systolic and diastolic contractile dysfunction. In addition, these mice show signs of eccentric left ventricular hypertrophy (increased HW/BW ratio and increased LV end-diastolic volume) and congestive heart failure (increased LW/BW ratios). Moreover, CaMKII expression was significantly increased, which is a hallmark for hypertrophy, contractile dysfunction and arrhythmias [8, 9, 31].

Since obstructive sleep apnea may result in the development of arterial hypertension with increased cardiac afterload, this may partly explain the cardiac phenotype of the present model. In fact, rodent OSA models showed OSA-dependent development of arterial hypertension [16, 32, 44].

However, all our mice subjected to PTFE injection had been healthy at baseline. Thus, all potential pathophysiologic changes that may have developed during the 8 weeks observation period, such as increased blood pressure, impaired myocardial contractility or impaired sleep

with chronic sympathetic stress, are secondary to the PTFE-induced intermittent airway obstruction during sleep. Consequently, all cardiovascular effects can be (directly or indirectly) attributed to OSA. This differentiates our model from obese and diabetic mouse models of sleep apnea, for instance, where OSA-independent comorbidities confound the experiments. Future studies using our model may address the relative contribution of the different pathophysiological alterations secondary to OSA individually.

The cardiovascular dysfunction developed by mice in our model is rather modest. A reduction of ejection fraction from 56.10±2.49% in control to 49.02±2.07% in PTFE mice may be detectable, but its pathophysiological relevance may be low. There are mouse models of systolic heart failure like transverse aortic constriction that would result in a much larger degree of systolic dysfunction [45]. Moreover, if compared to patients, the magnitude of ejection fraction observed in our PTFE mice would still be in the normal to subnormal range. On the other hand, we have observed many clinical features that can be found in patients with heart failure with <u>preserved</u> ejection fraction or patients with hypertension and hypertensive heart disease [46, 47]. Importantly, arterial hypertension and heart failure with preserved ejection fraction are very common in patients with OSA and not only found in those patients at the far end of extremely severe intermittent airway obstruction [48–50].

The frequency of apnea events and the increase with PTFE-injection was rather modest in our model. In contrast, many other animal models exceed the severity of human OSA, partly because the consequences are to be detected within a few weeks [12, 16, 32, 33]. We have extended our observation period to a long duration of 8 weeks, other mouse models usually perform OSA protocols (e.g. CIH, tracheal occlusion) for about 3–5 weeks [12, 16, 20, 32, 35, 36]. We did this to model the human situation more closely, where mild intermittent airway obstruction may result in the development of pathophysiological sequelae only after years. Despite this mild increase in intermittent airway obstruction, we show here that the frequency of apneas correlated significantly with the severity of contractile dysfunction and other features of the heart failure (heart and lung weight, Figs 4 and 5), suggesting a causal relationship. On the other hand, we cannot exclude that other factors following OSA that are not directly related to intermittent airway obstruction may also potentially contribute to the phenotype of these mice.

## Limitations

Although we conclusively report that PTFE-injected mice show features of OSA, we have not simultaneously monitored tidal volume and breathing effort. Therefore, we cannot exclude that some of the detected apneas may be a consequence of dysregulation by the central nervous system. However, we demonstrated a significant correlation between IFLs and apneas, suggesting that the majority of apneas observed in this model is obstructive in nature. In addition, the cut-off of 1 s for detection of apneas was arbitrarily chosen. We have not continuously monitored arterial oxygen concentration to show that 1 s apneas would result in arterial desaturation. Intriguingly, KDM6A has recently been reported to be a novel and high sensitive hypoxic marker that is decreased upon hypoxia [27]. Importantly, we have observed that mRNA expression of KDM6A was significantly decreased in ventricular myocardium of PTFE-treated mice, indicating that hypoxia may be present. Since there was also a significant negative correlation between the tongue diameter and KDM6A expression, upper airway obstruction via tongue enlargement seems to potentially induce apnea-dependent hypoxemia. Our findings were confirmed by mRNA expression analysis of hypoxia-inducible factor 1α (HIF1α) that was upregulated in the ventricular myocardium of PTFE-treated mice, further suggesting that arterial hypoxia may be present [22, 24–26]. HIF1α has been shown to be a valuable marker

for hypoxemia [22, 25, 26]. It enhances the transcription of either adaptive or deleterious genes, depending on the intensity and duration of hypoxemia [25, 26]. In addition to the regulation of HIF1α by protein stabilization [51], several *in vivo* studies showed increased levels of HIF1α mRNA in human, rats, and mice that were exposed to hypoxia [22, 24, 26]. Moreover, HIF1α mRNA expression is less vulnerable to fluctuation due to the critical timing and method of the euthanasia compared to HIF1α protein [22, 24–26, 51]. Interestingly, HIF1α mRNA expression has been shown to be already increased after 30 min of hypoxia (at 7% $O_2$) *in vivo* in mice [24]. Moreover, the total increase of HIF1α mRNA expression to about 2.5-fold was also comparable to a model of CIH [26].

We have not directly monitored sleep cycles by electroencephalography, which would have required a substantial additional methodological effort. On the other hand, performing sleep apnea monitoring without electroencephalography during the usual rodent sleep cycle (day-time, e.g. 9 AM to 5 PM) has been shown to be feasible [20]. Additionally, we have shown here that IFLs do not occur with a uniform distribution across the monitoring interval. Instead, they form clusters when mice were supposed to sleep, while no IFL aggregates and only a very low frequency of IFLs was observed at night-time when mice were awake. Nevertheless, further investigations are required to directly correlate IFLs and apneas with sleep.

## Conclusions

In conclusion, we describe here the first mouse model showing spontaneous IFLs, obstructive apneas and hypoxia by tongue enlargement due to PTFE injection in the absence of co-morbidities. These mice develop systolic and diastolic dysfunction and increased CaMKII expression. This model circumvents many problems of current animal models of sleep apnea. It is feasible and readily available to many researchers, which renders it an ideal tool to investigate mechanisms of OSA especially in the context of genetically modified mice, which may help identify novel treatment strategies that are urgently needed.

## Supporting information

**S1 Fig. Study flowchart.** Study flowchart showing the allocation of 59 mice in total. 31 mice were subjected to tongue enlargement by PTFE and 28 littermates were used as control animals. 16 mice (5 control vs. 11 PTFE mice) were used for some proof of principle experiments that are part of the Supporting information. Since they are not part of the main manuscript, they are not shown in this study flowchart.
(TIF)

**S2 Fig. Increased cross-sectional tongue area and sustained OSA at 8 weeks after PTFE injection.** a) Original ultrasound image of a murine tongue in transversal plane before and after (PTFE) injection (left panel). Interestingly, we observed a strong increase in the lateral tongue diameter. The mean data for 5 animals is shown in right panel. PTFE injection resulted in a significant increase in both lateral (transversal plane) and dorso-ventral (sagittal plane) tongue diameters. Cross-sectional tongue area was calculated by lateral diameter*dorso-ventral diameter*0.25*π estimating an elliptical shape of the tongue. b) After PTFE injection, the proportion of mice showing an abnormally increased apnea frequency (cut-off 14.75 apneas/h) was significantly increased. c) Importantly, frequencies of apneas, IFLs, and IFL aggregates remained stable for the whole 8-week observation period (N = 6). *—P<0.05 vs. pre injection (a) or control (b), one-way repeated measures ANOVA with Holm-Sidak's post-hoc correction (a+c) and Chi-square test (b).
(TIF)

**S3 Fig. Increased IFL frequency in mice after PTFE injection.** a) PTFE-injected mice showed a significant increase in the absolute frequency of inspiratory flow limitations (IFLs/h). b) In accordance, we observed a significantly increased percentage of flow limited breaths in PTFE mice that also correlated significantly positive with the frequency of apneas. $^*$—P<0.05, Student's t-test and linear regression analysis, as appropriate.
(TIF)

**S4 Fig. Breathing characteristics in conscious mice are unaltered by PTFE injection.** a) Average breath frequency, and the number of IFL and apneas were calculated for 30 min intervals during a 22 h observation period (from 10 p.m. to 8 p.m. the other day) in a PTFE-treated mouse. Time of activity of conscious mice can be easily discriminated from sleep time by monitoring average breathing frequency. An increased number of IFLs was typically accompanied by a concomitant increase in the number of apneas only during sleep time. b) Original box flow recordings of a control (upper panel) and a PTFE-treated mouse (lower panel) measured by whole-body plethysmography in conscious mice. Both mice showed a similar breathing pattern, indicating that tongue enlargement due to PTFE injection does not induce upper airway obstruction in conscious mice. c) Mean data for IFL and apnea frequency during activity time. In conscious mice (activity time at night), a negligible number of apneas and IFL could be detected with no difference between PTFE-treated and control animals. Kruskal-Wallis test with Dunn's post-hoc correction.
(TIF)

**S5 Fig. HIF1α mRNA expression is upregulated in PTFE mice.** HIF1α mRNA expression was analyzed by qPCR (normalized to β-actin) from hearts. a) Scatter plots of HIF1α mRNA expression in control (N = 5) and PTFE-treated (N = 8) animals (left panel). There was a significant upregulation of HIF1α mRNA expression after PTFE treatment. The level of HIF1α mRNA expression correlated significantly with the frequency of apneas (right panel). b) Interestingly, the tongue diameter correlated significantly positive with the HIF1α expression, indicating hypoxemia due to PTFE-dependent tongue enlargement. $^*$—P<0.05, Mann-Whitney test and linear regression analysis, as appropriate.
(TIF)

**S1 Raw images. Gels.**
(PDF)

**S1 Video. Tongue injection procedure.**
(MOV)

## Acknowledgments

We greatly appreciate the expert technical assistance of Gabriela Pietrzyk, Thomas Sowa, and Felicia Radtke.

## Author Contributions

**Conceptualization:** Simon Lebek, Christian Schach, Lars Siegfried Maier, Michael Arzt, Stefan Wagner.

**Data curation:** Simon Lebek, Philipp Hegner, Christian Schach, Kathrin Reuthner, Maria Tafelmeier.

**Formal analysis:** Simon Lebek, Philipp Hegner, Kathrin Reuthner, Maria Tafelmeier, Stefan Wagner.

**Funding acquisition:** Lars Siegfried Maier, Michael Arzt, Stefan Wagner.

**Investigation:** Simon Lebek, Philipp Hegner, Christian Schach, Kathrin Reuthner, Maria Tafelmeier, Stefan Wagner.

**Methodology:** Simon Lebek, Philipp Hegner, Christian Schach, Kathrin Reuthner, Maria Tafelmeier, Stefan Wagner.

**Project administration:** Lars Siegfried Maier, Stefan Wagner.

**Resources:** Christian Schach, Lars Siegfried Maier, Michael Arzt, Stefan Wagner.

**Software:** Lars Siegfried Maier, Stefan Wagner.

**Supervision:** Lars Siegfried Maier, Michael Arzt, Stefan Wagner.

**Validation:** Michael Arzt, Stefan Wagner.

**Visualization:** Simon Lebek.

**Writing – original draft:** Simon Lebek.

**Writing – review & editing:** Philipp Hegner, Maria Tafelmeier, Lars Siegfried Maier, Michael Arzt, Stefan Wagner.

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
