## [Decision Letter · Decision Letter 0]

14 Oct 2020

PONE-D-20-28615

Obstructive sleep apnea by bulking agent-induced tongue enlargement results in left ventricular contractile dysfunction

PLOS ONE

Dear Dr. Wagner,

Thank you for submitting your manuscript to PLOS ONE. After careful consideration, we feel that it has merit but does not fully meet PLOS ONE’s publication criteria as it currently stands. Therefore, we invite you to submit a revised version of the manuscript that addresses the points raised during the review process.

We look forward to receiving your revised manuscript.

Kind regards,

Michael Bader

Academic Editor

PLOS ONE

Journal Requirements:

2. Please modify the title to ensure that it is meeting PLOS’ guidelines (https://journals.plos.org/plosone/s/submission-guidelines#loc-title). In particular, the title should be "specific, descriptive, concise, and comprehensible to readers outside the field" and in this case we feel that the animal model used should be included. Please amend both the title on the online submission form (via Edit Submission) and the title in the manuscript so that they are identical.

3. Please clarify whether the method of euthanasia used was cervical dislocation.

'I have read the journal's policy and the authors of this manuscript have the following competing interests: MA received grant support from ResMed, the ResMed Foundation, and Philips Respironics as well as lecture and consulting fees from ResMed, Philips Respironics, Boehringer-Ingelheim, NRI, Novartis and Bresotec. There are no other competing interests to declare.'

a. Please confirm that this does not alter your adherence to all PLOS ONE policies on sharing data and materials, by including the following statement: "This does not alter our adherence to  PLOS ONE policies on sharing data and materials.” (as detailed online in our guide for authors http://journals.plos.org/plosone/s/competing-interests).  If there are restrictions on sharing of data and/or materials, please state these.

Please note that we cannot proceed with consideration of your article until this information has been declared.

6. Please include captions for your Supporting Information files at the end of your manuscript, and update any in-text citations to match accordingly. Please see our Supporting Information guidelines for more information: http://journals.plos.org/plosone/s/supporting-information

Reviewers' comments:

Reviewer's Responses to Questions

**Comments to the Author**

1. Is the manuscript technically sound, and do the data support the conclusions?

Reviewer #1: Partly

Reviewer #2: Partly

2. Has the statistical analysis been performed appropriately and rigorously? 

Reviewer #1: Yes

Reviewer #2: Yes

3. Have the authors made all data underlying the findings in their manuscript fully available?

Reviewer #1: Yes

Reviewer #2: Yes

4. Is the manuscript presented in an intelligible fashion and written in standard English?

Reviewer #1: Yes

Reviewer #2: Yes

5. Review Comments to the Author

Reviewer #1: The main aim of the present manuscript by Lebek and collaborators is the description of a new model of obstructive sleep apnea (OSA) in rodents by injecting polytetrafluoroethylene in the base of their tongue, thebe increasing its size. By mean of plethysmography, the authors found that apneas are doubled in OSA mice. The number of apneas correlate with systolic and diastolic function, and hypoxia molecular markers. Tongue size remained stable for 8 weeks.

The manuscript is well written and, in general, easy to follow.

Other OSA animal models have been published and are discussed. However, some others needing few manipulation or setup are not mentioned (eg, Rubies et al. Sci Rep. 2019;9:11443). Amongst similar models, the authors acknowledge injection of a variety of inert or biological substances in the tongue or larynges of large (monkey) and small (rabbit, rat) animals. The main advance of the present model is its use in mice and the possibility to use transgenic animals. Whether the model developed in rat (hyaluronate injection) could be used in mice has not been studied.

I have several major comments.

One of my most important concerns is whether apneas/flow limitations occur during sleep only. The authors’ state so on the basis of a subjective assessment of flow patterns during light and dark periods, but objective data is warranted. Flow and apnea recordings during awake periods (night time, darkness) in both groups, and formal comparisons, are warranted.

Some of the authors claims are not sufficiently supported and need more supporting data. The authors state that the model was very effective. According to figure 2b, there was a large overlap in the number of apneas in control and OSA mice. In which percentage of OSA animals the number of apneas was higher than normal (ie, “normal” could be defined as the mean+2SD apneas in the control group)?

The authors also claim that a single injection results in sustained airway obstruction at 8 weeks. Nevertheless, the authors show that tongue size remains stable from baseline to 8 weeks. One may argue that, while tongue size remains stable, mice keep growing and, thereby, the relative obstruction (and thereby, apnea effectiveness) is lower at the 8-week timepoint. The authors do only demonstrate a significant increase in apnea frequency a the 2-week timepoint, but not at the 8-week timepoint.

In addition to those animals that had to be sacrificed and data on normal weight gain, was any evidence of stress or pain evident?

The number of induced apneas is rather modest: on average, less than doubles the number of apneas. In contrast, other animal models and OSA in human increase the number of apneas by several-fold. However, the authors show a remarkable cardiovascular affectation, including systolic dysfunction. Could the authors discuss?

Could other factors play a role? The authors claim that the cardiovascular effects of the present OSA model may not be caused by comorbidities. However, OSA promotes an increase in bloop pressure, and resistant hypertension (Tietjens et al. J Am Heart Assoc 2019;8: e010440), but blood pressure is not tested.

Statistical analyses are, in general, appropriate. Was normality assessed? Were paired t-test or repeated measures ANOVA performed with those with >1 measurement per animal?

Considering that this paper is mainly describing a new method, it may be informative providing a recording of tongue injections.

Were mice sacrificed during the light or dark period?

How was tongue echography performed?

Were ventricular samples obtained from the left or right ventricle? Could the authors show the full WB lane?

Both in the introduction and the discussion, the authors claim that CPAP may be harmful to OSA patients on the basis of SERVE-HF trial. However, the authors statement might be misleading and should be corrected. The Cowie et al. trial did include patients with predominant central apneas, in contrast to an OSA population.

Reviewer #2: I do have some comments related to some of sections of the manuscript:

*Title: The title of the paper does reflect what was done but does not seem to follow the overall rationale of the manuscript. I would suggest editing the title to match what was done with greater accuracy. Make clear to the reader that this was done in mice, with the aim to present a novel animal model of OSA.

*Methods

1- How was the 100 uL amount determined? Were other amounts tested previously? If yes, please include this in the paper and how the investigators reached a final decision to use 100 uL.

2- The plethysmography was done during the daytime. I realize that mice are nocturnal animals, and I would stress in the methods sections that the recordings were done during the sleep cycle. On a related note, what happens to levels of hypoxia etc when they are awake?

3- Based on figure 1 a , it seems very arbitrary how tongue volume was measured. My questions to the authors are : How were the images standardized? Was a specific magnification used? Were all these measurements done by the same investigator? Was there a calibration? Any Kappa statistics to be reported? Are there any fluorescence techniques to show the areas of injection? This should be added to the paper? Finally, the imaging seems to be in 2D, while you are referring to tongue volume. If a 3D measurement was done, more detail is needed about how the different planes of space were oriented, etc.

The whole paper is based on the increase tongue volume, it would be beneficial to have more details about how the tongue volume was assessed.

*Discussion/Conclusions

Based on the authors findings on the increased heart and lung weight, significant increases in CaMKII and KDM6A, it is quite clear to me that these mice developed cardiovascular morbidity from the intervention. That being the case, I disagree that this is a purely a model of OSA in mice. It can be argued that this is on the far end of extremely severe OSA, which may be encountered in heart failure patients. I would suggest acknowledging these findings and adapting the text to reflect that. Still, these data suggest a novel approach for a future valid OSA model, however it poses the question whether this animal model would be able to represent the burden of solely due to OSA.

6. PLOS authors have the option to publish the peer review history of their article (what does this mean?). If published, this will include your full peer review and any attached files.

Reviewer #1: No

Reviewer #2: No

---

## [Author Response · Author response to Decision Letter 0]

27 Nov 2020

Response to the Editors

• A rebuttal letter that responds to each point raised by a You should upload this letter as a separate file labeled 'Response to Reviewers'.

Response: We have resubmitted all required documents.

Response: We have revised the manuscript according to PLOS ONE's style requirements.

Response: We have revised the title page according to the journal guidelines.

2. Please modify the title to ensure that it is meeting PLOS’ guidelines (https://journals.plos.org/plosone/s/submission-guidelines#loc-title). In particular, the title should be "specific, descriptive, concise, and comprehensible to readers outside the field" and in this case we feel that the animal model used should be included. Please amend both the title on the online submission form (via Edit Submission) and the title in the manuscript so that they are identical.

Response: We appreciate this important comment and have revised the title accordingly. It now reads “A novel mouse model of obstructive sleep apnea by bulking agent-induced tongue enlargement results in left ventricular contractile dysfunction”. Moreover, we have also revised the short title that now reads “A novel mouse model of sleep apnea with contractile dysfunction”.

3. Please clarify whether the method of euthanasia used was cervical dislocation.

Response: Indeed, we used cervical dislocation for euthanasia. We therefore state on page 6 in lines 85-87:

“All animals were euthanized by cervical dislocation during the light period, i.e. regular sleep time of the animals (usually in the afternoon).”

Response: We now provide the full unedited gels for Fig 6 in the Supporting Information, labeled “S1 Gels. Raw images.”, according to the journal guidelines. There are no other blot/gel image data related to this manuscript. In addition, all figures adhere to the journal guidelines.

Response: We now provide the full unedited gels underlying Fig 6 in the Supporting Information, labeled “S1 Gels. Raw images.”. There are no other blot/gel image data related to this manuscript.

'I have read the journal's policy and the authors of this manuscript have the following competing interests: MA received grant support from ResMed, the ResMed Foundation, and Philips Respironics as well as lecture and consulting fees from ResMed, Philips Respironics, Boehringer-Ingelheim, NRI, Novartis and Bresotec. There are no other competing interests to declare.'

a. Please confirm that this does not alter your adherence to all PLOS ONE policies on sharing data and materials, by including the following statement: "This does not alter our adherence to PLOS ONE policies on sharing data and materials.” (as detailed online in our guide for authors http://journals.plos.org/plosone/s/competing-interests). If there are restrictions on sharing of data and/or materials, please state these.

Please note that we cannot proceed with consideration of your article until this information has been declared.

Response: We confirm that our competing interests do not alter our adherence to all PLOS ONE policies on sharing data and materials. Accordingly, we have added to the Competing Interests on page 29 in lines 629-630:

“This does not alter our adherence to PLOS ONE policies on sharing data and materials.”

Response: We have included our updated Competing Interests statement in our cover letter.

6. Please include captions for your Supporting Information files at the end of your manuscript, and update any in-text citations to match accordingly. Please see our Supporting Information guidelines for more information: http://journals.plos.org/plosone/s/supporting-information

Response: We have included in the revised manuscript captions for our Supporting Information files at the end of our manuscript on pages 41-42 and have updated all in-text citations according to the journal guidelines.

Comments to the Author

1. Is the manuscript technically sound, and do the data support the conclusions?

Reviewer #1: Partly

Reviewer #2: Partly

2. Has the statistical analysis been performed appropriately and rigorously? 

Reviewer #1: Yes

Reviewer #2: Yes

3. Have the authors made all data underlying the findings in their manuscript fully available?

Reviewer #1: Yes

Reviewer #2: Yes

4. Is the manuscript presented in an intelligible fashion and written in standard English?

Reviewer #1: Yes

Reviewer #2: Yes

5. Review Comments to the Author

 

Response to the Reviewers

Reviewer #1: The main aim of the present manuscript by Lebek and collaborators is the description of a new model of obstructive sleep apnea (OSA) in rodents by injecting polytetrafluoroethylene in the base of their tongue, thebe increasing its size. By mean of plethysmography, the authors found that apneas are doubled in OSA mice. The number of apneas correlate with systolic and diastolic function, and hypoxia molecular markers. Tongue size remained stable for 8 weeks.

The manuscript is well written and, in general, easy to follow.

Other OSA animal models have been published and are discussed. However, some others needing few manipulation or setup are not mentioned (eg, Rubies et al. Sci Rep. 2019;9:11443). Amongst similar models, the authors acknowledge injection of a variety of inert or biological substances in the tongue or larynges of large (monkey) and small (rabbit, rat) animals. The main advance of the present model is its use in mice and the possibility to use transgenic animals. Whether the model developed in rat (hyaluronate injection) could be used in mice has not been studied.

Response: We thank the reviewer for these important comments. Indeed, several animal models of OSA have been published. We have discussed these different models and approaches without being exhaustive. We agree with the reviewer that the model of Rubies et al. is interesting and needs to be included in our discussion. We have therefore added the following sentence on page 21 lines 435-438 to the discussion section of the revised manuscript.

“Another rat model of OSA mimicking airway obstruction was first described by Farré et al. [35]. Rats were placed awake in a setup with two chambers split by a latex neck collar and airway obstruction was induced by interruption of bias flow in the head chamber [35,36].”

Moreover, we now also discuss that the previously published rat model of upper airway obstruction by hyaluronate injection may also be applicable to mice. You can read on page 24 in lines 500-505:

“Additionally, polyacrylamide and sodium hyaluronate injection into the palate of rabbits and rats, respectively, has been shown to induce obstructive sleep apnea by upper airway obstruction [42,43]. Unfortunately, the availability of appropriate knock-out or transgene models is very limited in both rabbits and rats. In this context, future studies investigating sodium hyaluronate injection into the palate of mice may lead to another promising animal model of OSA.”

I have several major comments.

One of my most important concerns is whether apneas/flow limitations occur during sleep only. The authors’ state so on the basis of a subjective assessment of flow patterns during light and dark periods, but objective data is warranted. Flow and apnea recordings during awake periods (night time, darkness) in both groups, and formal comparisons, are warranted.

Response: We appreciate this important comment. Indeed, it is essential to delineate whether inspiratory flow limitations (IFLs) and apneas occur during sleep only. To account for this, we have performed novel experiments for the revised version of the manuscript by measuring breathing patterns (whole-body plethysmography) during night-time, darkness, when mice are awake. To also investigate the shift from awake breathing to sleep-related breathing we have, furthermore, extended the recording period to 22 hours (from 10 p.m. to 8. p.m. the other day). As obvious from the original recording (novel figure a) in S4 Fig) IFLs and apneas do not occur during night-time (10 p.m. – 7 a.m. the other day) in awake PFTE-treated mice. In accordance, no airway obstruction was observed during this period (b) in S4 Fig) In sharp contrast, as evident from the original recording (novel figure a) in S4 Fig), the same PTFE-treated mouse showed a substantial increase in IFL/apnea frequency during the light period (7 a.m. – 8 p.m.), when mice are sleeping. Interestingly, the shift from awake to sleep time may be even inferred from the strong decrease in breathing frequency occurring around 7 a.m. and remaining low during daytime. The lack of IFL and apneas occurrence during night-time/awake time was also evident from mean data analyzed from 5 mice (novel figure c) in S4 Fig). During night-time, we observed a very low frequency of IFLs and apneas in PTFE-treated mice that was comparable to control mice (P=1.00 for both, c) in S4 Fig). Importantly, no IFL aggregates were observed in awake mice. 

We have added the novel data to the results section of the revised manuscript on page 15 in lines 310-316:

“When mice were awake (during night-time, 10 p.m. – 7 a.m.), control and PTFE mice showed a similar breathing pattern with a negligible number of IFLs and no IFL aggregates indicating that PTFE injection into the tongue does not induce a fixed upper airway obstruction (S4 Fig). In conscious mice, mean IFL frequency (/h) was 1.98±0.52 vs. 2.49±0.51 (PTFE vs. control; N=5 for both; P=1.00; c) in S4 Fig). Also, mean apnea frequency (/h) was very low and similar in awake PTFE and control mice (1.95±1.01 vs. 2.38±1.19; N=5 for both; P=1.00; c) in S4 Fig).”

Moreover, we discuss these findings in the revised version of the manuscript on page 23-24 in lines 487-495:

“Importantly, PTFE injection into the tongue does not lead to a fixed upper airway obstruction (S4 Fig). In sleeping mice, only about 0.50% of all breaths in PTFE mice were flow limited (b) in S3 Fig). In accordance with the intermittent nature of obstructive breathing abnormalities, the PTFE-induced IFLs and apneas occurred in clusters and only in sleeping mice (a) in S4 Fig). In contrast, awake PTFE mice exhibit a regular breathing pattern without airway obstruction (b) in S4 Fig). Moreover, we observed no IFL aggregates and similar frequencies of IFLs and apneas after PTFE treatment during awake periods (c) in S4 Fig). Therefore, hypoxia is likely not present in awake mice.”

Although we can now demonstrate that the PTFE-induced apneas and IFLs occur mainly during the murine sleep time, we cannot correlate the apneas and IFLs with definite sleep, e.g. using EEG. We have accounted for this aspect in the limitations section on page 28 in lines 595-603:

“We have not directly monitored sleep cycles by electroencephalography, which would have required a substantial additional methodological effort. On the other hand, performing sleep apnea monitoring without electroencephalography during the usual rodent sleep cycle (day-time, e.g. 9 AM to 5 PM) has been shown to be feasible [20]. Additionally, we have shown here that IFLs do not occur with an uniform distribution across the monitoring interval. Instead, they form clusters when mice were supposed to sleep, while no IFL aggregates and only a very low frequency of IFLs was observed at night-time when mice were awake. Nevertheless, further investigations are required to directly correlate IFLs and apneas with sleep.”

Some of the authors claims are not sufficiently supported and need more supporting data. The authors state that the model was very effective. According to figure 2b, there was a large overlap in the number of apneas in control and OSA mice. In which percentage of OSA animals the number of apneas was higher than normal (ie, “normal” could be defined as the mean+2SD apneas in the control group)?

Response: We thank the reviewer for this important comment. Indeed, there was a substantial variation in the in the number of apneas in control and PTFE mice, which is not unusual for this type of measurement. As suggested by the reviewer, we have performed a novel analysis for the revised version of the manuscript: by using the mean±2SD of the control group, a cut-off of 14.75 apneas/h was used to discriminate between normal or abnormal increased apnea frequency. Interestingly, 8 out of 25 PTFE mice (32%) but only 2 out of 28 control mice showed an abnormal increased apnea frequency (P=0.02, Chi-square test). We believe that this substantial and significant increase in the number of abnormally breathing mice after PTFE treatment is convincing. We have added the novel analysis to the result section (page 15, lines 299-304) and b) in novel S2 Fig:

“Moreover, the proportion of mice showing an abnormally increased apnea frequency above the cut-off of 14.75 apneas/h (mean apnea frequency of control mice + 2 standard deviations) was significantly increased in PTFE-injected mice (b) in S2 Fig). Interestingly, 8 out of 25 PTFE mice but only 2 out of 28 control mice showed an abnormally increased apnea frequency (P=0.02; b) in S2 Fig).”

The authors also claim that a single injection results in sustained airway obstruction at 8 weeks. Nevertheless, the authors show that tongue size remains stable from baseline to 8 weeks. One may argue that, while tongue size remains stable, mice keep growing and, thereby, the relative obstruction (and thereby, apnea effectiveness) is lower at the 8-week timepoint. The authors do only demonstrate a significant increase in apnea frequency a the 2-week timepoint, but not at the 8-week timepoint.

Response: We thank the reviewer for this important comment. Indeed, it is essential to ensure that IFL and apnea frequency remains stable for the whole 8-week follow-up period. We can now show in the revised version of the manuscript novel data comparing frequencies of IFLs, IFL aggregates, and apneas at the 2-week with the 8-week timepoint (novel figure c) in S2 Fig). Importantly, all parameters (IFLs, IFL aggregates, apneas) remain stable for the whole observation period and no significant difference was observed. We have added this finding to the results section on page 16 in lines 327-331:

“Importantly, the intermittent airway obstruction in sleeping mice remained stable for the whole observation period. Compared to the 2-week timepoint, frequencies of apneas (P=0.71), IFLs (P=0.38), and IFL aggregates (P=0.95) were similar at 8 weeks after PTFE injection (N=6 for all; c) in S2 Fig).”

Moreover, we have discussed this observation on page 23 in lines 485-487:

“According to the sustained increase in tongue diameter, we could demonstrate that the increased frequencies of IFLs and apneas remained stable for the whole 8-week observation period (c) in S2 Fig).”

In addition, to those animals that had to be sacrificed and data on normal weight gain, was any evidence of stress or pain evident?

Response: We thank the reviewer for this important comment. In order to respect animals’ wellbeing and to avoid animal suffering, we performed everyday visual inspection of every mouse in the study. In particular, we analyzed their skin, food intake, movements and interaction with other mice. If a mouse showed an abnormal behavior, we immediately sacrificed the animal (only 6 mice had to be sacrificed (S1 Fig)). All the other mice (25/31) showed no evidence of stress or pain and could be monitored for the whole observation period. To account for this, we added to the method section on page 8 in lines 127-134:

“From 31 mice treated with PTFE, 6 mice had to be killed within 72 h because of surgery-related complications (e.g. bleeding into the tongue, extensive tongue enlargement or infection). In order to respect animals’ wellbeing and to avoid animal suffering, we performed everyday visual inspection of every mouse. In particular, we analyzed their skin, food intake, movements and interaction with other mice. If a mouse showed an abnormal behavior, we immediately sacrificed the animal (6 mice had to be sacrificed (S1 Fig)). All the other mice (25/31) showed no evidence of stress or pain and could be monitored for the whole observation period of 8 weeks.”

The number of induced apneas is rather modest: on average, less than doubles the number of apneas. In contrast, other animal models and OSA in human increase the number of apneas by several-fold. However, the authors show a remarkable cardiovascular affectation, including systolic dysfunction. Could the authors discuss?

Response: This is an important comment. We have discussed the severity of intermittent airway obstruction in comparison to other animal models and its impact on pathophysiology on page 26, lines 552-565 of the discussion section of the revised manuscript:

“The frequency of apnea events and the increase with PTFE-injection was rather modest in our model. In contrast, many other animal models exceed the severity of human OSA, partly because the consequences are to be detected within a few weeks [12,16,33,32]. We have extended our observation period to a long duration of 8 weeks, other mouse models usually perform OSA protocols (e.g. CIH, tracheal occlusion) for about 3-5 weeks [12,16,35,36,20,32]. We did this to model the human situation more closely, where mild intermittent airway obstruction may result in the development of pathophysiological sequelae only after years. Despite this mild increase in intermittent airway obstruction, we show here that the frequency of apneas correlated significantly with the severity of contractile dysfunction and other features of the heart failure (heart and lung weight, Figs 4 and 5), suggesting a causal relationship. On the other hand, we cannot exclude that other factors following OSA that are not directly related to intermittent airway obstruction may also potentially contribute to the phenotype of these mice.”

Could other factors play a role? The authors claim that the cardiovascular effects of the present OSA model may not be caused by comorbidities. However, OSA promotes an increase in bloop pressure, and resistant hypertension (Tietjens et al. J Am Heart Assoc 2019;8: e010440), but blood pressure is not tested.

Response: We thank the reviewer for this important comment. We agree with the reviewer that obstructive sleep apnea may result in the development of arterial hypertension with increased cardiac afterload. In fact, this may partly explain the cardiac phenotype of the present model. However, all animals subjected to PTFE injection had been healthy C57BL/6 mice at baseline. Thus, all potential pathophysiologic changes that may have developed during the 8 weeks observation period, let it be increased blood pressure, impaired myocardial contractility or impaired sleep with chronic sympathetic stress, are secondary to the PTFE-induced intermittent airway obstruction during sleep. Consequently, all cardiovascular effects can be (directly or indirectly) attributed to OSA. This differentiates our model from obese and diabetic mouse models of sleep apnea, for instance, where OSA-independent comorbidities confound the experiments. Future studies using our model may address the relative contribution of the different pathophysiological alterations secondary to OSA individually.

To account for this, we have added this aspect to the revised version of the manuscript on page 25 in lines 526-539:

“Since obstructive sleep apnea may result in the development of arterial hypertension with increased cardiac afterload, this may partly explain the cardiac phenotype of the present model. In fact, rodent OSA models showed OSA-dependent development of arterial hypertension [16,32,44].

However, all our mice subjected to PTFE injection had been healthy at baseline. Thus, all potential pathophysiologic changes that may have developed during the 8 weeks observation period, such as increased blood pressure, impaired myocardial contractility or impaired sleep with chronic sympathetic stress, are secondary to the PTFE-induced intermittent airway obstruction during sleep. Consequently, all cardiovascular effects can be (directly or indirectly) attributed to OSA. This differentiates our model from obese and diabetic mouse models of sleep apnea, for instance, where OSA-independent comorbidities confound the experiments. Future studies using our model may address the relative contribution of the different pathophysiological alterations secondary to OSA individually.”

Statistical analyses are, in general, appropriate. Was normality assessed? Were paired t-test or repeated measures ANOVA performed with those with >1 measurement per animal?

Response: We thank the reviewer for this important comment. In the revised version of the manuscript, we have now tested all data for normal distribution using Shapiro-Wilk normality test. Consequently, we used a parametric or a non-parametric test, depending on whether a variable was normally distributed or not, respectively. Moreover, one may argue that in some figures longitudinal observations are presented and paired t-tests or repeated measures ANOVA may be considered. All these data sets are discussed here: 

1) in b) in Fig 1, data for 8 weeks was only available for a subgroup of mice, which hampers the use of a repeated measures ANOVA. 

2) in c) in Fig 1, a two factor repeated measures study design was present. As recommended by the reviewer we have now performed a mixed-effects model analysis with Holm-Sidak’s post-hoc correction.

3) In novel S2 Fig, longitudinal data is compared, wherefore we used one-way repeated measures ANOVA with Holm-Sidak’s post-hoc correction.

We have therefore revised the methods section on page 13 in lines 255-272 now describing this new statistical analysis:

“All measurements and experiments were performed and analyzed blinded to the treatment group (control or PTFE) and to frequency of apneas. Experimental data are presented as means ± standard error of the mean (SEM). All statistical analyses were based on the number of mice and normal distribution was assessed by Shapiro-Wilk normality test. Parametric or non-parametric tests were applied to test for significant differences, depending on whether a variable was normally distributed or not. Parametric and non-parametric tests used for the comparison of two groups were Student’s t and Mann-Whitney test, respectively. Ordinary one-way ANOVA with Holm-Sidak’s post-hoc correction and Kruskal-Wallis test with Dunn’s post-hoc correction were used for comparisons of more than two groups that were either normally or not normally distributed, respectively. One-way repeated measures ANOVA with Holm-Sidak’s post-hoc correction was used for the comparison of paired data that was normally distributed. If more than two groups and two different factors were compared in a repeated measures design, mixed-effects model analysis with Holm-Sidak’s post-hoc correction was used. Chi-square test was used for the comparison of categorial data. The tests above as well as linear regression analyses were used in GraphPad Prism 8 to test for significance, as appropriate. Two-sided P-values below 0.05 were considered as statistically significant.”

Considering that this paper is mainly describing a new method, it may be informative providing a recording of tongue injections.

Response: We appreciate this important comment. We have performed a video recording of the surgical procedure of PTFE tongue injection. It can be found in the revised version of the manuscript as Supporting Information named “S1 Video”. Tongue injection procedure”.

Were mice sacrificed during the light or dark period?

Response: This is an important comment. All mice were sacrificed during the light period, i.e. regular sleep time of the animals. We have added this information on page 6 in lines 85-87:

“All animals were euthanized by cervical dislocation during the light period, i.e. regular sleep time of the animals (usually in the afternoon).”

How was tongue echography performed?

Response: We thank the reviewer for this comment and have now added the novel paragraph “Sonographic measurement of tongue diameter” to the revised method section on pages 8-9 in lines 136-153:

“Tongue size was measured by ultrasound during the PTFE injection procedure. Mice were placed in supine position onto a heating plate. The tongue was gripped with a tiny crocodile clip. Ultrasound gel was placed onto the murine throat, mandible and mouth, but not on nostrils to keep mice breathing. Thereafter, a 30 MHz center frequency transducer (Vevo3100 system from VisualSonics, Toronto, Canada) was placed at median position of the murine throat to measure the dorso-ventral tongue diameter in sagittal plane. For some recordings, the ultrasound head was rotated clockwise by 90° to also measure the lateral tongue diameters in the transversal plane (a) in S2 Fig). Recordings were acquired with 56 frames/s (gain 30 dB). For optimal magnification, acquisition was performed with 10.00 mm depth and 15.36 mm width. We used the presetting of VisualSonics; thus, no calibration was required. By carefully stirring the tongue via the crocodile clip and comparing tongue movements with the other pharyngeal structures under sonographic recording, tongue surface was easily discriminated from surrounding tissue and tongue diameter was assessed. Similar measurements were done before and after PTFE injection in a standardized manner. All measurements were done by the same investigator; therefore, Kappa statistics cannot be reported. We did not use any fluorescence techniques to identify the area of injection.”

Were ventricular samples obtained from the left or right ventricle? Could the authors show the full WB lane?

Response: We thank the reviewer for this comment. In order to save enough material for protein analyses, we used the whole left and right ventricle for homogenization. We have added this information to the method section (page 12, line 237). In addition, we now show the full Western blot gels of CaMKII and GAPDH as Supporting Information “S1 Gels. Raw images”.

Both in the introduction and the discussion, the authors claim that CPAP may be harmful to OSA patients on the basis of SERVE-HF trial. However, the authors statement might be misleading and should be corrected. The Cowie et al. trial did include patients with predominant central apneas, in contrast to an OSA population.

Response: We thank the reviewer for this comment and apologize this misunderstanding. We have specified the statements in the introduction and in the discussion accordingly.

It now reads on page 4 in lines 48-50 in the introduction:

“While treatment with ventilation-therapy may reduce apnea events, not all patients can tolerate it [6] and this treatment may even be harmful for selected patients (e.g. for patients with predominant central apneas) [7].”

We have also specified the statement in the discussion on page 20 in lines 405-409:

“To date, the treatment of OSA is mainly limited to continuous positive airway pressure (CPAP), but acceptance of CPAP in patients with low symptom burden is limited [6] and treatment with positive pressure ventilation (adaptive servo-ventilation) may be even harmful for specific patient populations (e.g. for patients with predominant central apneas) [7].”

Reviewer #2: I do have some comments related to some of sections of the manuscript:

*Title: The title of the paper does reflect what was done but does not seem to follow the overall rationale of the manuscript. I would suggest editing the title to match what was done with greater accuracy. Make clear to the reader that this was done in mice, with the aim to present a novel animal model of OSA.

Response: We appreciate this important comment and have revised the title accordingly. It now reads “A novel mouse model of obstructive sleep apnea by bulking agent-induced tongue enlargement results in left ventricular contractile dysfunction”. Moreover, we have also revised the short title that now reads “A novel mouse model of sleep apnea with contractile dysfunction”.

*Methods

1- How was the 100 uL amount determined? Were other amounts tested previously? If yes, please include this in the paper and how the investigators reached a final decision to use 100 uL.

Response: This is an important comment. Our approach was based on the findings of Brennick et al., who had measured pharyngeal structures of New Zealand Obese mice (NZO) with spontaneous OSA using MRI (Brennick MJ, Pack AI, Ko K, Kim E, Pickup S, Maislin G, et al. Altered upper airway and soft tissue structures in the New Zealand Obese mouse. Am J Respir Crit Care Med. 2009; 179: 158–169). Interestingly, they report that NZO mice (aged 23 weeks, mean body weight 35.7 g) showed a significantly increased mean tongue volume to about 137 µl (compared to 104 µl in control animals). This corresponds to a mean increase of 33 µl tongue volume. Since the tongue volume is the most important determinant of pharyngeal airway size for OSA (Schwab RJ, Pasirstein M, Pierson R, Mackley A, Hachadoorian R, Arens R, et al. Identification of upper airway anatomic risk factors for obstructive sleep apnea with volumetric magnetic resonance imaging. Am J Respir Crit Care Med. 2003; 168: 522–530), we aimed to increase the tongue volume of our mice to a similar extent by PTFE injection into the base of the tongue. We used younger mice (mean body weight 27.7 g, only about 70% of the body weight compared to Brennick et al.) to enable the 8-week follow-up observation period. Thus, we anticipated that an increase of about 20-25 µl tongue volume would result in a similar airway obstruction. PTFE is a solid substance (density 2.1 g/ml). 50 mg of PTFE was diluted to 100 µl (50% w/v) with glycerol (Sigma Aldrich). 100 µl of this dilution contains 24 µl pure PTFE, which almost exactly matches the aimed increase in tongue volume. Larger PTFE injection volumes were investigated in some test mice, but periprocedural mortality exceeded. Since we were not interested in less upper airway obstruction, we have not studied lower injection volumes.

We have added this information to the method section of the revised version of the manuscript on pages 6-7 in lines 94-110:

“Our approach was based on the findings of Brennick et al., who had measured pharyngeal structures of New Zealand Obese mice (NZO) with spontaneous OSA using MRI [14]. Interestingly, they report that NZO mice (aged 23 weeks, mean body weight 35.7 g) showed a significantly increased mean tongue volume to about 137 µl (compared to 104 µl in control animals). This corresponds to a mean increase of 33 µl tongue volume. Since the tongue volume is the most important determinant of pharyngeal airway size for OSA [19], we aimed to increase the tongue volume of our mice to a similar extent by PTFE injection into the base of the tongue. We used younger mice (mean body weight 27.7 g, only about 70% of the body weight compared to Brennick et al. [14]) to enable the 8-week follow-up observation period. Thus, we anticipated that an increase of about 20-25 µl tongue volume would result in a similar airway obstruction. PTFE is a solid substance (density 2.1 g/ml). 50 mg of PTFE was diluted to 100 µl (50% w/v) with glycerol (Sigma Aldrich). 100 µl of this dilution contains 24 µl pure PTFE, which almost exactly matches the aimed increase in tongue volume. Larger injection volumes were investigated in some test mice, but periprocedural mortality exceeded. Since we were not interested in less upper airway obstruction, we have not studied lower injection volumes.”

2- The plethysmography was done during the daytime. I realize that mice are nocturnal animals, and I would stress in the methods sections that the recordings were done during the sleep cycle. On a related note, what happens to levels of hypoxia etc when they are awake?

Response: We thank the reviewer for this helpful comment. We now explain in the method section on page 9 in lines 164-167 that recordings were conducted during the murine sleep cycle:

“Since mice are nocturnal animals, continuous recordings (sampling frequency 1 kHz) were done for 8 h during day-time, the interval with the highest frequency and duration of sleep periods complying with the murine sleep cycle [20].”

Moreover, we have added novel data to the revised version of the manuscript investigating breathing characteristics at night-time (activity time), when mice are awake (S4 Fig). Intriguingly, awake mice showed no IFL aggregates and only very low frequencies of IFLs and apneas with no differences between the control and PTFE-treated animals. Consistently, the level of intermittent hypoxia should be negligible in PTFE-treated awake mice and comparable to control mice.

We have added this aspect to the revised discussion section on pages 23-24 in lines 487-495:

“Importantly, PTFE injection into the tongue does not lead to a fixed upper airway obstruction (S4 Fig). In sleeping mice, only about 0.50% of all breaths in PTFE mice were flow limited (b) in S3 Fig). In accordance with the intermittent nature of obstructive breathing abnormalities, the PTFE-induced IFLs and apneas occurred in clusters and only in sleeping mice (a) in S4 Fig). In contrast, awake PTFE mice exhibit a regular breathing pattern without airway obstruction (b) in S4 Fig). Moreover, we observed no IFL aggregates and similar frequencies of IFLs and apneas after PTFE treatment during awake periods (c) in S4 Fig). Therefore, hypoxia is likely not present in awake mice.”

3- Based on figure 1 a, it seems very arbitrary how tongue volume was measured. My questions to the authors are: How were the images standardized? Was a specific magnification used? Were all these measurements done by the same investigator? Was there a calibration? Any Kappa statistics to be reported? Are there any fluorescence techniques to show the areas of injection? This should be added to the paper? Finally, the imaging seems to be in 2D, while you are referring to tongue volume. If a 3D measurement was done, more detail is needed about how the different planes of space were oriented, etc.

The whole paper is based on the increase tongue volume, it would be beneficial to have more details about how the tongue volume was assessed.

Response: We appreciate this important comment and agree with the reviewer that measurement of the tongue size should be described more in detail. Therefore, we have added a novel paragraph “Sonographic measurement of tongue diameter” to the methods section of the revised manuscript.

On pages 8-9, lines 136-153 the text reads as following:

“Tongue size was measured by ultrasound during the PTFE injection procedure. Mice were placed in supine position onto a heating plate. The tongue was gripped with a tiny crocodile clip. Ultrasound gel was placed onto the murine throat, mandible and mouth, but not on nostrils to keep mice breathing. Thereafter, a 30 MHz center frequency transducer (Vevo3100 system from VisualSonics, Toronto, Canada) was placed at median position of the murine throat to measure the dorso-ventral tongue diameter in sagittal plane. For some recordings, the ultrasound head was rotated clockwise by 90° to also measure the lateral tongue diameters in the transversal plane (a) in S2 Fig). Recordings were acquired with 56 frames/s (gain 30 dB). For optimal magnification, acquisition was performed with 10.00 mm depth and 15.36 mm width. We used the presetting of VisualSonics; thus, no calibration was required. By carefully stirring the tongue via the crocodile clip and comparing tongue movements with the other pharyngeal structures under sonographic recording, tongue surface was easily discriminated from surrounding tissue and tongue diameter was assessed. Similar measurements were done before and after PTFE injection in a standardized manner. All measurements were done by the same investigator; therefore, Kappa statistics cannot be reported. We did not use any fluorescence techniques to identify the area of injection.”

We agree with the reviewer that it would have been better to measure tongue volume instead of diameters only. However, a precise measurement of tongue volume by 3D ultrasound would require the measurement of tongue length in addition to dorso-ventral and lateral diameter. Since the tip of the tongue was gripped with a tiny crocodile clip, the ultrasound head would not be able to reach it. Therefore, precise measurement of tongue length is not possible. Nevertheless, for the revised version of the manuscript we have now performed novel experiments to measure dorso-ventral and lateral tongue diameter in the same animal by rotating the ultrasound head 90 degrees to measure the transversal plane. In panel a) in S2 Fig we now report dorso-ventral and lateral tongue diameters and calculated cross-sectional area following PTFE injection (novel panel a) in S2 Fig). Intriguingly, both tongue diameters increased in parallel and to a similar extent resulting in a significant increase of cross-sectional tongue area (a) in S2 Fig).

We have added these novel findings to the results section on page 14 in lines 278-280:

“Interestingly, we observed a similar increase in transversal tongue diameter leading to a homogenous increase of cross-sectional tongue area from (in mm²) 9.23±0.41 to 19.90±0.86 (N=5; P<0.001; a) in S2 Fig).”

Moreover, to avoid misunderstanding, we have revised the entire manuscript and clarified our terminology: we now use either tongue diameter, cross-sectional area or volume, as appropriate.

*Discussion/Conclusions

Based on the authors findings on the increased heart and lung weight, significant increases in CaMKII and KDM6A, it is quite clear to me that these mice developed cardiovascular morbidity from the intervention. That being the case, I disagree that this is a purely a model of OSA in mice. It can be argued that this is on the far end of extremely severe OSA, which may be encountered in heart failure patients. I would suggest acknowledging these findings and adapting the text to reflect that. Still, these data suggest a novel approach for a future valid OSA model, however it poses the question whether this animal model would be able to represent the burden of solely due to OSA.

Response: We appreciate this important comment. However, we disagree on the statement that our model would be at the far end of extremely severe OSA. 

On page 26, lines 540-551 of the discussion section of the revised manuscript we explain this matter. The text reads as following:

“The cardiovascular dysfunction developed by mice in our model is rather modest. A reduction of ejection fraction from 56.10±2.49% in control to 49.02±2.07% in PTFE mice may be detectable, but its pathophysiological relevance may be low. There are mouse models of systolic heart failure like transverse aortic constriction that would result in a much larger degree of systolic dysfunction [45]. Moreover, if compared to patients, the magnitude of ejection fraction observed in our PTFE mice would still be in the normal to subnormal range. On the other hand, we have observed many clinical features that can be found in patients with heart failure with preserved ejection fraction or patients with hypertension and hypertensive heart disease [46,47]. Importantly, arterial hypertension and heart failure with preserved ejection fraction are very common in patients with OSA and not only found in those patients at the far end of extremely severe intermittent airway obstruction [48–50].“

In addition, we state on page 25, lines 526-539:

“Since obstructive sleep apnea may result in the development of arterial hypertension with increased cardiac afterload, this may partly explain the cardiac phenotype of the present model. In fact, rodent OSA models showed OSA-dependent development of arterial hypertension [16,32,44].

However, all our mice subjected to PTFE injection had been healthy at baseline. Thus, all potential pathophysiologic changes that may have developed during the 8 weeks observation period, such as increased blood pressure, impaired myocardial contractility or impaired sleep with chronic sympathetic stress, are secondary to the PTFE-induced intermittent airway obstruction during sleep. Consequently, all cardiovascular effects can be (directly or indirectly) attributed to OSA. This differentiates our model from obese and diabetic mouse models of sleep apnea, for instance, where OSA-independent comorbidities confound the experiments. Future studies using our model may address the relative contribution of the different pathophysiological alterations secondary to OSA individually.”

With respect to the severity of intermittent airway obstruction we also state on page 26, lines 552-565:

“The frequency of apnea events and the increase with PTFE-injection was rather modest in our model. In contrast, many other animal models exceed the severity of human OSA, partly because the consequences are to be detected within a few weeks [12,16,33,32]. We have extended our observation period to a long duration of 8 weeks, other mouse models usually perform OSA protocols (e.g. CIH, tracheal occlusion) for about 3-5 weeks [12,16,35,36,20,32]. We did this to model the human situation more closely, where mild intermittent airway obstruction may result in the development of pathophysiological sequelae only after years. Despite this mild increase in intermittent airway obstruction, we show here that the frequency of apneas correlated significantly with the severity of contractile dysfunction and other features of the heart failure (heart and lung weight, Figs 4 and 5), suggesting a causal relationship. On the other hand, we cannot exclude that other factors following OSA that are not directly related to intermittent airway obstruction may also potentially contribute to the phenotype of these mice.”

6. PLOS authors have the option to publish the peer review history of their article (what does this mean?). If published, this will include your full peer review and any attached files.

Response: We agree publishing the complete peer review history.

Response: We have used PACE digital diagnostic tool to ensure that all figures meet PLOS requirements.

---

## [Decision Letter · Decision Letter 1]

30 Nov 2020

A novel mouse model of obstructive sleep apnea by bulking agent-induced tongue enlargement results in left ventricular contractile dysfunction

PONE-D-20-28615R1

Dear Dr. Wagner,

We’re pleased to inform you that your manuscript has been judged scientifically suitable for publication and will be formally accepted for publication once it meets all outstanding technical requirements.

Kind regards,

Michael Bader

Academic Editor

PLOS ONE

Additional Editor Comments (optional):

Reviewers' comments:

Reviewer's Responses to Questions

**Comments to the Author**

1. If the authors have adequately addressed your comments raised in a previous round of review and you feel that this manuscript is now acceptable for publication, you may indicate that here to bypass the “Comments to the Author” section, enter your conflict of interest statement in the “Confidential to Editor” section, and submit your "Accept" recommendation.

Reviewer #1: All comments have been addressed

2. Is the manuscript technically sound, and do the data support the conclusions?

Reviewer #1: Yes

3. Has the statistical analysis been performed appropriately and rigorously? 

Reviewer #1: Yes

4. Have the authors made all data underlying the findings in their manuscript fully available?

Reviewer #1: Yes

5. Is the manuscript presented in an intelligible fashion and written in standard English?

Reviewer #1: Yes

6. Review Comments to the Author

Reviewer #1: The authors have provided a comprehensive and excellent reply to all my comments. They are to be commended for their effort. I have no further comments.

7. PLOS authors have the option to publish the peer review history of their article (what does this mean?). If published, this will include your full peer review and any attached files.

Reviewer #1: No

---

## [Editor Report · Acceptance letter]

2 Dec 2020

PONE-D-20-28615R1 

A novel mouse model of obstructive sleep apnea by bulking agent-induced tongue enlargement results in left ventricular contractile dysfunction 

Dear Dr. Wagner:

I'm pleased to inform you that your manuscript has been deemed suitable for publication in PLOS ONE. Congratulations! Your manuscript is now with our production department. 

Kind regards, 

on behalf of

Prof. Michael Bader 

Academic Editor

PLOS ONE